# A model framework to retrieve thermodynamic and kinetic properties of organic aerosol from composition-resolved thermal desorption measurements

Siegfried Schobesberger[1,2], Emma L. D'Ambro[3], Felipe D. Lopez-Hilfiker[1,a], Claudia Mohr[1,4], Joel A.
Thornton[1]

[1]Department of Atmospheric Sciences, University of Washington, Seattle, Washington 98195, USA.
[2]Department of Applied Physics, University of Eastern Finland, Kuopio, 70211, Finland.
[3]Department of Chemistry, University of Washington, Seattle, Washington 98195, USA.
[4]Department of Environmental Science and Analytical Chemistry, Stockholm University, Stockholm, 10691, Sweden.

[a] *now at:* Tofwerk AG, Thun, 3600, Switzerland.

*Correspondence to*: Siegfried Schobesberger (siegfried.schobesberger@uef.fi)

**Abstract.** Chemical ionization mass spectrometer (CIMS) techniques have been developed that allow for quantitative and composition-resolved measurements of organic compounds as they desorb from secondary organic aerosol (SOA) particles, in particular during their heat-induced evaporation. One such technique employs the Filter Inlet for Gases and AEROsol (FIGAERO). Here, we present a newly-developed model framework with the main aim of reproducing FIGAERO-CIMS thermograms: signal vs. ramped desorption temperature. The model simulates the desorption of organic compounds during controlled heating of filter-sampled SOA particles, plus the subsequent transport of these compounds through the FIGAERO manifold into an iodide-CIMS. Desorption is described by a modified Hertz-Knudsen equation and controlled chiefly by the temperature-dependent saturation concentration $C^*$, mass accommodation (evaporation) coefficient, and particle surface area. Subsequent transport is governed by interactions with filter and manifold surfaces. Reversible accretion reactions (oligomer formation and decomposition) and thermal decomposition are formally described following the Arrhenius relation. We use calibration experiments for tuning instrument-specific parameters, and then apply the model to a test case: measurements of SOA generated from dark ozonolysis of α-pinene. We then discuss the ability of the model to describe thermograms from simple calibration experiments and from complex SOA, and the associated implications for the chemical and physical properties of the SOA. For major individual compositions observed in our SOA test case (#C = 8 to 10), the thermogram peaks can typically be described by assigning $C^*_{25°C}$ values in the range 0.05 to 5 µg m$^{-3}$, leaving the larger, high-temperature fractions (>50%) of the thermograms to be described by thermal decomposition, with dissociation rates on the order of ~ 1 hr$^{-1}$ at 25 °C. We conclude with specific experimental designs to better constrain instrumental model parameters and to aid in resolving remaining ambiguities in the interpretation of more complex SOA thermogram behaviors. The model allows retrieval of quantitative volatility and mass transport information from FIGAERO thermograms, and for examining the effects of various environmental or chemical conditions on such properties.

# 1 Introduction

A large fraction of organic aerosol (OA) mass and cloud condensation nuclei (CCN) in the continental boundary layer are typically produced by condensation or reactive uptake of organic vapors to form secondary organic aerosol, SOA, (e.g., Hallquist et al., 2009; Riipinen et al., 2012). Some atmospheric models describe growth and evaporation of SOA by an absorptive partitioning of organic vapors between the gas and the particle phase, which is primarily controlled by the volatility of the involved compounds, usually expressed as either saturation vapor pressure ($P^*$) or saturation vapor concentration ($C^*$) (Pankow, 1994; Donahue et al., 2011). Important simplifying assumptions typically made are that the system is in equilibrium and that the condensed organic phase can be thought of as an ideal liquid solution. However, such descriptions of SOA dynamics have proven inadequate for predicting SOA mass abundance and properties (e.g., Heald et al., 2005; Dzepina et al., 2009; Virtanen et al., 2010). Correspondingly, equilibrium-partitioning models also fail in describing certain observations of SOA growth and evaporation, both for laboratory-generated and ambient SOA. Specifically, observed aerosol formation kinetics infer sets of volatilities for the involved vapors that predict a much faster evaporation of the SOA than is observed when the condensable vapors in the gas-phase are diluted or removed (Vaden et al., 2011; Yli-Juuti et al., 2017). Similar conclusions have been made from heat-induced aerosol evaporation experiments, where observed OA evaporation indicates a major fraction of material with lower volatility than indicated by OA growth or corresponding composition of evaporated compounds (Stanier et al., 2007; Cappa and Jimenez, 2010; Lopez-Hilfiker et al., 2015; Lopez-Hilfiker et al., 2016b).

Several hypotheses have been proposed to explain the inability of absorptive partitioning models to replicate such observations.

(a) Descriptions of gas-phase radical chemistry are inaccurate, e.g. missing an important role of highly oxygenated (peroxy-)functionalized molecules (e.g., Ehn et al., 2014) that could form a major component of SOA with extremely low vapor pressure. Note that such compounds may be relatively thermodynamically unstable (Krapf et al., 2016).

(b) Assumptions of particle phase state are invalid. On one hand, the organic constituents may not be ideally mixed (Robinson et al., 2015; Zuend and Seinfeld, 2012). Also, several types of ambient biogenic SOA particles have been shown to be not liquid, at least at certain humidity ranges, but to rather adopt an amorphous semisolid (i.e. glassy) state (Virtanen et al., 2010; Pajunoja et al., 2016). Such non-idealities can affect the effective volatility of SOA, e.g. via introducing limitations to in-particle diffusion (Cappa and Wilson, 2011; Shiraiwa and Seinfeld, 2012; Saleh et al., 2013; Renbaum-Wolff et al., 2013).

(c) Multiphase accretion chemistry is not adequately described. For instance, the formation of oligomers from oxygenated organics in the particle phase has been shown to occur in SOA in various conditions, in particular in laboratory experiments (e.g., Kalberer et al., 2004; Surratt et al., 2006; Romonosky et al., 2017). Amongst other forms of multiphase chemistry, it is a form of aerosol aging and has been observed to occur on timescales of hours in laboratory setups (e.g., Baltensperger et al., 2005). It lowers particle volatility and is likely occurring in ambient SOA as well

(Rudich et al., 2007; Kourtchev et al., 2016). Note that such chemistry may also constitute mechanisms that underlie the issues raised under (b) (Stroeve, 1975; Pfrang et al., 2011). Indeed, recent experimental and modeling studies have corroborated an important role of oligomerization in determining SOA behavior. Best model agreements with chamber studies have been reported when assuming rapid oligomerization reactions (within minutes) upon SOA formation, and as a consequence, oligomer decomposition may indeed control SOA evaporation rates (Trump and Donahue, 2014; Roldin et al., 2014; Kolesar et al., 2015b).

In recent years, various mass spectrometric techniques have been developed to provide relatively non-invasive methods of measuring aerosol molecular composition, such that particle-phase oligomers can be characterized. Some methods accomplish that via liquid extraction, either offline (e.g., Roach et al., 2010; Laskin et al., 2013; Beck and Hoffmann, 2016) or online (e.g., Doezema et al., 2012). Other methods first heat the aerosol particles, so that individual (organic) molecules thermally desorb from the condensed phase; the abundance and composition of these molecules can then be measured by chemical ionization or proton-transfer reaction mass spectrometry (CIMS, PTR-MS) (e.g., Smith et al., 2004; Hearn and Smith, 2004; Gkatzelis et al., 2018). Ideally, these techniques are coupled to mass spectrometers with high sensitivity, mass accuracy and resolving power, e.g. time-of-flight (TOF) mass spectrometers (Zhao et al., 2014; Eichler et al., 2015). A sub-class of these techniques heats the aerosol particles in a step-wise or continuously ramped manner, such that the thermal desorption behavior (thermograms) of the aerosol in general, and also of the individual desorbing molecules are measured simultaneously with the molecular formulas (e.g., Holzinger et al., 2010; Yatavelli et al., 2012). Measurements by one of the most recent developments within this sub-class of techniques are the main subject of this work, namely the Filter Inlet for Gases and AEROsol (FIGAERO; Lopez-Hilfiker et al., 2014) that is coupled to a high-resolution TOF CIMS (Lee et al., 2014).

Measurements by FIGAERO of ambient SOA, as well as of SOA generated in the lab following α-pinene oxidation, have shown that a substantial fraction of organic material is desorbing only at much higher temperatures than expected for the volatilities as known or expected from the detected compositions of the desorbing molecules (Lopez-Hilfiker et al., 2015; Lopez-Hilfiker et al., 2016b). This behavior was attributed to thermal decomposition of low volatility components (either individual molecules or oligomeric material) upon heating. These findings support the hypothesis that oligomer formation and decomposition may play an important role in determining SOA properties, in particular SOA evaporation upon heating or removal of condensing vapor, but the exact molecular-scale/chemical mechanisms at play have remained unknown. Speculations have included ubiquitous peroxides (cf., Docherty et al., 2005) with breakage of the O–O bond upon heating, networks of H-bridge bonds in the SOA matrix that are stronger or denser than for pure compounds or ideal mixtures, and oligomeric structures initially in thermodynamic equilibrium with monomers and thus dissociating during heating to re-achieve equilibrium (Lopez-Hilfiker et al., 2015). Consequently, we are using a broad and inclusive definition of the term "oligomer" in this study, referring to any physical entity that is essentially non-volatile but incorporates and/or releases generally more volatile molecules (the latter in particular upon heating). I.e., our definition is considerably more universal than the frequent use of the term as referring specifically to covalently bound large molecular weight molecules.

Recently, there have been additional notable attempts in improving our understanding of which physical and chemical aspects of OA control the results obtained by FIGAERO-CIMS measurements, both in terms of overall particle properties and of composition-specific chemistry. Stark et al. (2017) present detailed comparisons between the results obtained from different FIGAERO versions and similar thermal desorption techniques, as well as between alternative data analysis approaches. Their conclusions are consistent with those of Lopez-Hilfiker et al. (2015; 2016b) (see also above). Huang et al. (2018) performed a so far unique set of chamber experiments by employing a FIGAERO to study α-pinene SOA at various humidity and temperature conditions, in particular with chamber temperatures as low as 223 K. They conclude that particle viscosity likely affects the apparent volatilities obtained by FIGAERO, and that viscosity may be linked to particle water uptake and oligomer content. However, still lacking from such studies is a first-principles based model of the thermal desorption processes occurring in the FIGAERO, which would allow systematic interpretations of the measured thermograms in terms of instrumental conditions and SOA properties such as the effective volatility distribution of components.

For this study, we have developed a detailed model of the temperature-controlled evaporation of OA in the FIGAERO. The goal is to allow for a deeper understanding of which properties of OA, overall and component-specific, determine the shapes of the thermograms and their respective desorption temperatures obtained by the FIGAERO measurements. We first describe the model concepts and then the application to various thermogram calibration experiments using known compounds as a way to optimize instrumental parameters that affect mass transfer of evaporated material to the CIMS detector. We then apply the model to thermograms of SOA generated in a chamber from the oxidation of the monoterpene α-pinene to demonstrate the type of fundamental properties that can be retrieved from such comparisons, such as the reaction rates and energies that govern oligomer formation and decomposition.

## 2 Experimental Methods

### 2.1 FIGAERO-CIMS

The primary experimental data used for this research was obtained by an iodide-adduct high-resolution time-of-flight chemical ionization mass spectrometer (CIMS), as described in previous works (e.g. Lee et al., 2014), with a FIGAERO inlet. By means of CIMS, gas-phase compounds are primarily detected when they form adducts with iodide anions while inside an ion-molecule reaction region (IMR) at a pressure of 100 mbar. The analyte-reagent clusters pass through a differentially pumped interface to a time-of-flight mass spectrometer ($10^{-6}$ mbar), where their exact mass-to-charge ratio is measured and hence their elemental composition determined. This method is most sensitive to oxidized compounds, including a wide range of VOC oxidation products, in particular to organics featuring –O–H moieties (Iyer et al., 2016).

The FIGAERO inlet permits the investigation of particle-phase composition by collecting aerosol particles on a filter and then heating the filter while sampling the desorbing compounds (Fig. 1). Schematics of the FIGAERO setup for aerosol collection and for evaporation and sampling, and a detailed characterization can be found in Lopez-Hilfiker et al. (2014). We

briefly summarize key components here for understanding certain aspects of the model. Aerosol is first collected on a polytetrafluoroethylene (PTFE) filter (Zefluor PTFE membrane, 2 μm pore size, 25 mm diameter, Pall), usually over a period of the order of ~ 40 minutes. Then, the filter is moved ~ 5cm over the directly adjacent CIMS inlet where a flow of 2 slpm of ultra-pure $N_2$ passes through the filter, and then into the IMR by means of an orifice that allows for a pressure drop from atmospheric pressure (at the filter) to 100 mbar (in the pumped IMR). The $N_2$ flow is heated at a constant ramp rate from room temperature to 200 °C, typically at 10 °C $min^{-1}$, and is then kept at 200 °C for an additional period of time, typically 50 min, that is sufficient for a vast majority of detected material to desorb from the filter. The CIMS samples continuously during the full desorption period, yielding a thermogram (signal from desorbing composition vs. ramped temperature) for each desorbing composition, with the measured signal presumably directly proportional to the composition's rate of desorption. Note that the CIMS can measure only elemental compositions, i.e. molecular formulas (we are using these two terms interchangeably in this work). Consequently, the identities of the specific compounds remain ambiguous in general.

## 2.2 Filter properties

The collection efficiency of the used PTFE membrane filters is >98% for all particle sizes (Zíková et al., 2015). The filter material consists of two layers: a thicker mat consisting of a PTFE web of bonded PTFE fibers (oriented upstream in our measurements), and a thinner microporous PTFE membrane consisting of fibrils interconnected via nodes (oriented downstream). We were not able to obtain more detailed specific product information from the manufacturer, but general information on the filter materials is available in patents (e.g., US5366631 and US4187390). This information suggests that the web's fibers have diameters between 12 and 30 μm. We measured a filter mat thickness of 188 (± 6) μm. Assuming a material density of 2.2 g $cm^{-3}$, its measured weight inferred a solidity (= ratio of the volume of the layer's solid material to the layer's total volume) of 0.43 (± 0.02). The membrane's fibrils are suggested to have diameters ranging from 0.5 to 100 nm, and our measurements indicated a membrane thickness of 14 (± 3) μm and a solidity of 0.14 (± 0.03).

## 2.3 Experiment setups

In this study, we mostly rely on previously published results from thermogram calibration experiments (Lopez-Hilfiker et al., 2014; Lopez-Hilfiker et al., 2016b) and from SOA formation experiments conducted during an intensive measurement campaign at the Pacific Northwest National Laboratory's (PNNL) 10.6 $m^3$ environmental chamber. Thermogram calibrations were performed by using a micro-syringe to manually deposit solutions containing calibrant compounds directly onto the FIGAERO filter (Lopez-Hilfiker et al., 2014). The setup of experiments at the PNNL laboratory chamber is described e.g. in Liu et al. (2016). Chamber data used here was obtained during a measurement campaign in summer 2015 that focused on investigating the chemistry of SOA formed from the oxidation of isoprene and monoterpenes. Results from a selection of FIGAERO-CIMS measurements from that campaign were recently published (D'Ambro et al., 2017).

For the experiments used here, relative humidity in the PNNL chamber was always 50%, and we used a monodisperse effloresced ammonium sulfate seed particle population of 50 nm in diameter. The chamber was operated in continuous flow reactor mode. As SOA precursor, α-pinene was injected at a constant rate to maintain a concentration of 10 ppbv in the absence of oxidation, and monitored by PTR-MS. The data used here were taken during conditions of dark ozonolysis of α-pinene at concentrations of $O_3$ at 84 ppbv and of α-pinene reacted at 6.7 ppbv. The studied SOA samples were taken once steady-state conditions had been established in the chamber, as determined by gas analyzers and aerosol mass concentrations measured by an Aerodyne aerosol mass spectrometer (AMS). Particle size distributions were monitored by a scanning mobility particle sizer (SMPS). The total volume put through the chamber was ~ 30-40 L min$^{-1}$, resulting in a theoretical residence time of 3 to 5 hours. Accordingly, steady state was typically achieved on a time scale of 1 day.

Typical SOA mass loadings in the chamber were 2 to 3 μg m$^{-3}$, and the FIGAERO achieved adequate filter loadings by sampling for 40 min periods at 2.5 L min$^{-1}$. Every 4[th] sample was a blank measurement, with an additional filter in the aerosol sampling line (Lopez-Hilfiker et al., 2014). Measurement results were continuously monitored and both filters were replaced when memory effects in the form of elevated backgrounds were noticed (on average once per week).

## 3 Model Description

The model developed for this study consists of a set of differential equations that describe mass transfer and evaporation from particle surfaces, optional temperature-dependent particle phase chemistry, such as accretion or thermal decomposition reactions, as well as partitioning to PTFE surfaces in the FIGAERO inlet. A schematic of the most important processes simulated by the model is shown in Fig. 1.

### 3.1 Evaporation rate

The central equation, which describes the desorption rate for a certain compound $i$ from a deposited aerosol particle, uses a modified form of the Hertz-Knudsen equation (Hertz, 1882; Cappa et al., 2007):

$$\frac{dN_i}{dt} = -\frac{1}{\sqrt{2\pi \cdot k_B \cdot m_i \cdot T}} \cdot P_i^*(T) \cdot \chi_i \cdot \alpha \cdot \Gamma(Kn) \cdot SA \tag{1}$$

Here, $N_i$ is the number of molecules of compound $i$ in the particle (condensed) phase, $k_B$ is the Boltzmann constant, $m_i$ is the compound's molecular mass, $T$ is the absolute temperature, $P_i^*$ is the compound's saturation vapor pressure, $\chi_i$ is a factor accounting for Raoult's Law, $\alpha$ is the evaporation coefficient, $\Gamma$ is a factor accounting for gas-phase diffusion limitations, and $SA$ is the surface area of the condensed-/gas-phase interface. The saturation vapor pressure $P_i^*$ is a strongly temperature-dependent function commonly described by the Clausius-Clapeyron relation and depending on the enthalpy of vaporization or sublimation, $\Delta H$:

$$P_i^* = P_{i,0}^* \cdot e^{-\frac{\Delta H}{R}\left(\frac{1}{T} - \frac{1}{T_0}\right)} \tag{2}$$

where $R$ is the universal gas constant and $P^{*}_{i,0}$ the saturation vapor pressure at room temperature $T_0$. The factor $\chi_i$ in Eq. (1) is the mass fraction of the compound in the condensed phase to take into account Raoult's Law (Donahue et al., 2006), i.e.,

$$\chi_i = \frac{m_i N_i}{\sum_i (m_i N_i)} \tag{3}$$

The evaporation coefficient $\alpha$ has a value between 0 and 1 and accounts for deviations of the theoretical maximum evaporation rate due to barriers to interfacial transfer, e.g. diffusion limitations within the condensed phase. The factor $\Gamma(Kn)$, also a value between 0 and 1, is a Fuchs-type function of the Knudsen number $Kn$

$$\Gamma = \frac{Kn^2 + Kn}{Kn^2 + 1.283 Kn + 0.75} \tag{4}$$

that takes into account resistance to evaporation due to gas-phase diffusion limitations. In the case of an ideally mixed or single-component liquid, $\alpha = 1$, and with a sufficiently small surface area, $Kn \gg 1$, thus $\Gamma = 1$.

The *SA* is based on an assumed spherical particle. All deposited material treated by the model is assumed to be present within that sphere, representing a single aerosol particle that presumably rests on the filter with negligible contact with the filter material (e.g. due to a contact angle of 180° or solid phase). For low-viscosity liquid particles, the actual *SA* could be smaller (e.g. deposition as high spherical cap) or larger (e.g. deposition as low spherical cap), resulting in the actual evaporation occurring slower (thermogram shifting to higher temperatures) or faster (thermogram shifting to lower temperatures), respectively.

For each model run only one particle is considered. Scaling up a single run's results, as we typically do, carries the assumption that all deposited particles are identical, and more importantly that all deposited particles are spatially separated from one another. For the chamber experiments here, SOA mass loadings were typically 2 μg m$^{-3}$, particles 100 nm or larger, and the collection time 45 min. In those conditions, < 1% of the FIGAERO filter area was loaded, on average, and the total mass loading was < 0.3 μg. Even if all SOA mass was deposited only on a smaller area corresponding to the inner cross-section of the sampling tube (ca. 4 mm inner diameter for PNNL experiments in 2015), local coverage would still be < 15%, so our assumptions are likely justified. It remains possible, however, that particles preferentially deposit in certain areas of the filter (i.e. on the microscopic scales of fibril nodes etc.). Huang et al. (2018) did report effects of filter mass loading on observed SOA thermograms, when loadings ranged from about 0.5 to 10 μg, indicating interactions between particles deposited on the filter. Their FIGAERO used a slightly different sampling geometry, which focused particles onto a smaller area of the filter, thus making matrix effects more likely. In any case though, the possibility of such effects, e.g. via reducing *SA*, should be kept in mind.

### 3.2 Vapor-surface interactions

Aerosol particles deposited in the FIGAERO are expected to be mostly located on or within the PTFE filter. Hence, we assume that evaporated molecules will not necessarily directly enter the CIMS, but that instead they first interact with PTFE surfaces, at least with the surfaces of the filter, possibly also with PTFE surfaces immediately surrounding the filter. Downstream from the filter, the desorbed molecules enter the IMR where they may again interact with PTFE surfaces,

namely the IMR walls, albeit at 100 mbar. As the residence time of air in the IMR is $\sim$ 30 ms, we expect interactions in the filter to be the dominant vapor-surface interactions because the filter provides a large total surface area, and the desorbed compounds need to pass through it prior to entering the mass spectrometer.

To account for these vapor-wall interactions, we adapt the approach used by Zhang et al. (2014) for modeling the wall losses of organic vapors in Teflon laboratory chambers:

$$\frac{dN_{i,w}}{dt} = k_{i,w,on} \cdot \left(-\frac{dN_i}{dt}\right) - k_{i,w,off} \cdot N_{i,w} \tag{5}$$

Here, $N_{i,w}$ is the number of molecules of compound $i$ on the wall, $k_{i,w,on}$ is the probability of ad- or ab-sorption into the wall, and $k_{i,w,off}$ is the rate constant for desorption off the wall. We set $k_{i,w,on}$ to 1, so $k_{i,w,off}$ is the quantity controlling the vapor-wall interaction. Assuming detailed balance and activity coefficients of unity,

$$k_{i,w,off} = \frac{k_{i,w,on}}{\tau} \cdot \frac{C_i^*(T)}{C_w} = \frac{C_i^*(T)}{\tau \cdot C_w} \tag{6}$$

where $C_w$ is an equivalent sorbing mass concentration represented by the walls, with the same units as the saturation vapor concentration of compounds $i$, $C_i^*$, for a treatment analogous to gas-particle partitioning. $C_w$ includes any possible non-unity vapor activity with respect to the wall, making it an effective concentration. Values for $C_w$ previously found for Teflon surfaces were 0.3 to 36 mg m$^{-3}$ for various organic vapors in a 8 m$^3$ fluorinated ethylene propylene (FEP) chamber (Matsunaga and Ziemann, 2010; Yeh and Ziemann, 2015; Krechmer et al., 2016) and 4 g m$^{-3}$ for ketones and alkenes in a 0.47 cm inner diameter perfluoroalkoxy alkane (PFA) tube (Pagonis et al., 2017). The time scale in Eq. (6), $\tau$, depends on the time scales of the processes involved in surface absorption. McMurry and Stolzenburg (1987) assumed diffusion-limited absorption determined by the characteristic times for diffusion to the surface ($\tau_{diff}$) and accommodation into it ($\tau_{ac}$), according to:

$$\tau_{diff} = \frac{d^2}{8D_g} \tag{7a}$$

$$\tau_{ac} = \frac{d}{2\alpha_W \bar{c}} \tag{7b}$$

Eq. (7a) has been applied to a laminar flow in a tube with inner diameter $d$, and $D_g$ being the gas-phase diffusion coefficient for the vapor in question. Eq. (7b) includes the compound's accommodation coefficient $\alpha_W$ and its mean thermal speed $\bar{c}$. Although our filter is not a tube, Eq. (7a) may serve to provide a potential upper limit time scale (if $\alpha_W$ is high) when using $d$ = 2 μm, the filter's nominal pore size, which yields $\tau_{diff} \approx 6.4 \times 10^{-8}$ s. For $\alpha_W = 1$, $\tau_{ac} = 5.5 \times 10^{-9}$ s, setting the lowest limit time scale. If $\alpha_W < 0.08$, $\tau_{ac}$ will be greater than $\tau_{diff}$ and thus the overall limiting time scale. Note that these times are much shorter than values typical for tubing or chambers, where $\tau_{diff}$ is typically limiting, and much longer. Conversely however, we expect $C_w$ to be much higher in our case than the literature values mentioned above, as it scales with the ratio of surface area to volume (Pagonis et al., 2017).

We can use observed timescales of specific compounds transiting the FIGAERO to obtain a robust estimate of the parameter product $\tau C_w$. We analyzed a variant of FIGAERO blanks, where an additional filter is placed upstream of the FIGAERO filter so that only some gas-phase compounds are present on the main filter through ad- or absorption (Lopez-Hilfiker et al.,

2014). In this variation of blank experiments, the clean $N_2$ flow for the subsequent desorption period was not heated, i.e. evaporation of the compounds desorbing from the FIGAERO filter occurred only at room temperature. Once exposed to pure $N_2$, the desorption rate for vapor $i$, $dN_i/dt$, should therefore be simply an exponential decay:

$$\frac{dN_i}{dt} = A \cdot e^{-\frac{C_{i,0}^*}{\tau \cdot C_w} \cdot t}$$

(8)

where $C_{i,0}^*$ is the saturation vapor concentration of compound $i$ at room temperature ($T_0$), and $A$ is a free parameter subject to the unknown amount of material deposited. Eq. (8) is not able to fit the experimental data (Fig. 2, brown line); instead a good fit is obtained by using two exponential terms (Fig. 2, green line):

$$\frac{dN_i}{dt} = A \cdot e^{-\frac{C_{i,0}^*}{\tau_1 \cdot C_{w1}} \cdot t} + B \cdot e^{-\frac{C_{i,0}^*}{\tau_2 \cdot C_{w2}} \cdot t}$$

(9)

where $B$ is a free parameter like $A$, and $C_{w1}$ and $C_{w2}$ represent two independent sets of PTFE surfaces, or two distinct ad-
/absorptive surface properties. We used a set of four isothermal desorption experiments and fit the decaying signals for two of the more abundant semi-volatile organics observed, potentially pinonic acid, measured as $C_{10}H_{16}O_3.\Gamma^-$, and pinic acid, measured as $C_9H_{14}O_4.\Gamma^-$. We obtained 8.8 ($\pm0.7$) mg m$^{-3}$ s for $\tau_1 C_{w1}$ and 150 ($\pm40$) mg m$^{-3}$ s for $\tau_2 C_{w2}$, together with a $C_{pinonic,0}^*$ of 510 ($\pm70$) µg m$^{-3}$ and a $C_{pinic,0}^*$ of 70 ($\pm14$) µg m$^{-3}$. Note that all these $C$-values could be multiplied by an arbitrary factor while maintaining the fits (Eq. (9)), but the ratio between the $C_{i,0}^*$ values would need to remain the same.
The actual saturation vapor concentrations for pinonic and pinic acid are not well known: literature reports range from 5.7 to 16000 µg m$^{-3}$ and from 2.6 to 1200 µg m$^{-3}$, respectively (Bilde and Pandis, 2001; Compernolle et al., 2011; Hartonen et al., 2013). Therefore, the suggested values of 510 µg m$^{-3}$ for $C_{pinonic,}^*$ and 70 µg m$^{-3}$ for $C_{pinic,0}^*$ are plausible in absolute terms, and their ratio of about one order of magnitude roughly corresponds to experimental findings.

We will see below that describing vapor-surface interactions using only $\tau_1 C_{w1}$ is sufficient for our applications of the model
to aerosol particle desorption. To examine the plausibility of this value, $\tau C_w = 8.8$ mg m$^{-3}$ s, we may use it to infer the filter's ratio internal surface area to volume. A reasonable range of $\tau$ is from 5.5 to 60 $\times$ 10$^{-9}$ s (see above). Scaling the corresponding range of $C_w$ (140 to 1600 kg m$^{-3}$) to the range of 0.3 to 36 mg m$^{-3}$, reported for FEP chambers of a surface-to-volume ratio of $\sim$ 3 m$^{-1}$, infers an internal surface area of 11 to 16000 m$^2$ per m$^2$ of filter area and µm of thickness. This range is plausible in comparison with the range of values suggested by available information about the filter membrane
(section 2.2): 4 to 900 µm$^{-1}$.

### 3.3 Model application to calibration experiments

As a test of model performance, we applied our model to calibration experiments that consisted of depositing a solution of mono-carboxylic acids directly onto the FIGAERO filter by means of a micro-syringe (Lopez-Hilfiker et al., 2014). The results are shown in Fig. 3. Better agreement between experimental and model results was achieved when using only $\tau_1 C_{w1}$
as the wall parameter (Fig. 3B) rather than $\tau_2 C_{w2}$ (Fig. 3C). Agreement was worse when neglecting vapor-surface interactions altogether (Fig. 3A) or when using both parameters. If both wall parameters were used in parallel (not shown), i.e. desorbing

material interacted either with surface *w1* or surface *w2*, the modeled thermograms would each display a double peak, which we did not observe for syringe experiments. Double peaks would also appear, in general, if we assumed that the surface interactions occurred in series, i.e. such that a fraction of the material that had interacted with surface *w1* also interacted with surface *w2*. And if that fraction was unity, the model result would be practically identical to Fig. 3C because $\tau_2 C_{w2} \gg \tau_1 C_{w1}$.

As a consequence, we used $\tau_1 C_{w1}$ (responsible for the fast decay in Fig. 2) as the single wall parameter in subsequent model runs. The requirement of using both $\tau_1 C_{w1}$ and $\tau_2 C_{w2}$ in analyzing the blank experiments above is possibly due to (slower) co-desorption of material from preceding experiments that had deposited onto surfaces that are less efficiently purged by the $N_2$ flow.

    As seen in Figs. 3A-C, the modeled temperatures of peak desorption agreed fairly well with the experimental results when

vapor-surface interactions after initial desorption are taken into account. However, the model performed poorly in reproducing the observed peak shapes, in particular the tails that became more substantial for less volatile compounds. The only way the observed peak shapes were simulated reasonably well, including the tails, was by assuming uneven heating of the deposited material. Under this assumption, only a part of the deposit was actually exposed to the nominal desorption temperature, whereas the remainder of the material was exposed to a certain fraction of that temperature at any given time.

Figure 4 illustrates this approach, with the resulting thermograms shown in Fig. 3D.

    As alternative attempts to broaden the modeled thermogram peaks, we tested a sequential vapor-surface interaction scheme, where desorbed molecules would interact with a series of surfaces at sequentially cooler temperatures, and the use of a distribution of $\tau C_w$ values. Both approaches generally enhanced the tailing of thermograms, but they failed to reproduce the observations of higher tails for less volatile compounds.

Finally, previous work demonstrated that the desorption characteristics largely do not depend on the method by which substance is delivered onto the filter. Lopez-Hilfiker et al. (2016b) used the FIGAERO to investigate the desorption of dipentaerithritol, deposited either via syringe in solution or via sampling aerosol produced by atomizing dipentaerithritol in water. The respective thermograms were similar, therefore, we believe that the model confirmation presented in this section, based on desorption of solution deposits, is applicable to desorption of aerosol deposits as well.

**3.4 Implementation of oligomerization reactions**

    To examine possible oligomerization reactions, or thermal decomposition more generally, we added two terms to Eq. (1) that describe the production and loss of compound *i* by the dissociation and formation of oligomers, respectively:

$$\frac{dN_i}{dt} = -\frac{1}{\sqrt{2\pi \cdot k_B \cdot m_i \cdot T}} \cdot SA \cdot \alpha \cdot \Gamma \cdot P_i^* \cdot \chi_i + k_d^i \cdot N_{i,g} - N_i \cdot \sum_j \left( K_g^{i,j} \cdot N_j \right) \qquad (10a)$$

    This procedure was inspired by Trump and Donahue (2014) and Kolesar et al. (2015a). Now, $N_i$ are the number of molecules

of compound *i* that are free to evaporate as dictated by the corresponding $P_i^*$ ("monomers"), whereas $N_{i,g}$ is the number of molecules bound in a state of lower volatility, e.g. in an oligomer, from which direct evaporation is assumed to be negligible. That is, we assume these oligomers are non-volatile:

$$\frac{dN_{i,g}}{dt} = -k_d^i \cdot N_{i,g} + N_i \cdot \sum_j \left( K_g^{i,j} \cdot N_j \right) \tag{10b}$$

The rate constants are $k_d^i$ for dissociation, and $K_g^{i,j}$ for oligomerization. The subscript $g$ is short for the deliberately non-descriptive "glued" or "gluing", as a reminder that the actual mechanism by which compound $i$ enters a state of lower volatility is not yet taken into account, for a lack of deeper understanding. This notation therefore reflects our broad definition of "oligomer" in this study as noted above (section 1). I.e., we refer to any physical entity that is itself non-volatile and able to incorporate and/or release compound $i$ as oligomer, as described by Eq. (10b).

The initial distribution of molecules of compound $i$ between $N_i$ and $N_{i,g}$ is calculated by assuming steady-state conditions at the initial temperature (= room temperature) and zero net evaporation, i.e. equal magnitudes of the 2nd and 3rd right-hand terms in Eq. (10a). Therefore, as the monomers start to undergo net evaporation upon removal of the gas-phase (typically coincident with the start of heating), oligomer dissociation (2nd term) will outpace oligomer formation (3rd term) until all molecules ($N_i + N_{i,g}$) have evaporated.

Note that by use of Eqs. (10a) and (10b) we do not track specific oligomers but rather the partitioning of compound $i$ between the two states (i.e., monomer vs. part of oligomer). Consequently, oligomer dissociation is independent of how compound $i$ entered the oligomer state. Also, in some cases, the "monomer" compound $i$ will itself be an oligomer, as for instance dimer-like compositions have been directly observed by FIGAERO-CIMS (Lopez-Hilfiker et al., 2015; Mohr et al., 2017; this study). In such a case, $N_{i,g}$ represents its involvement in yet larger complexes, whereas possible decomposition of the compound itself is not modelled.

Further simplification was needed for the oligomerization term in Eqs. (10a) and (10b), because for the majority of systems we investigate, we are unable to detect all relevant particle-phase constituents, let alone quantify their abundance with sufficient relative accuracy. In addition, it was desirable to reduce model complexity. Hence, we replaced the last term in Eqs. (10a) and (10b) with a pseudo-first order reaction term, so that Eq. (10a) becomes

$$\frac{dN_i}{dt} = -\frac{1}{\sqrt{2\pi \cdot k_B \cdot m_i \cdot T}} \cdot SA \cdot \alpha \cdot \Gamma \cdot P_i^* \cdot \chi_i + k_d^i \cdot N_{i,g} - k_g^i \cdot N_i \cdot \Phi \tag{11}$$

where $\Phi$ is the volume fraction of all organic compounds still present in the aerosol particle, ranging from one, at the beginning of desorption, to close to zero at the end.

In Eq. (11), both rate constants, $k_d^i$ for dissociation and $k_g^i$ for oligomerization, thus have units of $s^{-1}$. We treated these rates as temperature-dependent as in Arrhenius' equation, i.e. for each compound $i$,

$$k_d = k_{d,0} \cdot e^{-\frac{E_d}{R}\left(\frac{1}{T} - \frac{1}{T_0}\right)} = A_d \cdot e^{-\frac{E_d}{RT}} \tag{12}$$

and

$$k_g = k_{g,0} \cdot e^{-\frac{E_g}{R}\left(\frac{1}{T} - \frac{1}{T_0}\right)} = A_g \cdot e^{-\frac{E_g}{RT}} \tag{13}$$

where $A_d$ or $A_g$ would correspond to the pre-exponential factor in the traditional formulation of the Arrhenius equation. Oligomer formation and dissociation was thus described for each compound by four free parameters to be determined by fitting to experimental data: the rate constants at room temperature $k_{d,0}$ and $k_{g,0}$ and the respective activation energies $E_d$ and

$E_g$. The fraction of molecules initially present in the oligomer state was then simply $k_{g,0}/(k_{g,0}+k_{d,0})$. This fraction was used as an initial condition for $N_g$.

## 3.5 Further simplifications

Among the factors in Eq. (11), only $\chi_i$ and $\Phi$ are directly dependent on compounds other than compound $i$, while $SA$ and $\Gamma$ depend on the particle diameter $D_P$ and thus on $\Phi$. As such, both $\chi_i$ and $\Phi$ can contribute substantially to computational costs, and explicit calculation of all $N_i$ (i.e. all detected compounds) was feasible only for simple cases, such as certain calibration experiments. When applying our model to desorption data of OA components, we would instead reduce all OA mass to two compounds: the compound $i$ of interest and the sum of all other compounds. The latter sum is treated like a single composition by the model, and the respective model parameters may be unphysical, because the corresponding sum thermogram is a superposition of the thermogram signals of all individual compositions, which we know differ substantially in their respective volatilities. Nonetheless, the parameters are chosen such that the corresponding thermogram is adequately reproduced and thus allow us to use appropriate values for $\chi_i$, $D_P$ and $\Phi$ as functions of time. The model can then be run practically independently for each individual compound $i$, i.e. for reproducing each individual compound's thermogram as measured by FIGAERO.

For this case, the Raoult term $\chi_i$, particle diameter $D_P$ and the organic fraction remaining $\Phi$ are calculated specifically by

$$\chi_i(t) = \frac{m_i N_i(t)}{m_i N_i(t) + \bar{m} N_R(t)} \tag{14}$$

$$D_P(t) = \sqrt[3]{\frac{6}{\pi\rho}\left(m_i\big(N_i(t) + N_{i,g}(t)\big) + \bar{m}\big(N_R(t) + N_{R,g}(t)\big)\right)} \tag{15}$$

$$\Phi(t) = \left(\frac{D_P(t)}{D_{P,0}}\right)^3 \tag{16}$$

where the subscript $R$ denotes the sum of all organic compounds other than compound $i$, with a mean molecular mass of $\bar{m}$ and a molecular density of $\rho$. Where applicable, a refractive core (e.g. due to non-soluble inorganic seed particles) is taken into account through small modifications of Eqs. (15) and (16) employing basic geometry, as detailed in the supplementary material (Eqs. (S1) and (S2)).

Of course, this procedure yields only approximations, with the implicit assumptions: (a) that the FIGAERO detects all organic compounds and (b) that it does so with the same sensitivity for each compound. We know that FIGAERO coupled to iodide-CIMS appears to detect only about half of the organic material by mass under these assumptions, and that reported sensitivities generally vary widely (Lopez-Hilfiker et al., 2016b; Iyer et al., 2016). However, even if only half of the organic mass was accounted for, the directly introduced error would be comparable with an error in $C^*_i$ or $\alpha$ of up to about a factor of two, which will be a relatively small uncertainty given other ambiguities discussed below. Indeed, a recent study employed a calibration procedure for instrument sensitivity to most compositions and, within uncertainties, obtained mass closure with independent AMS or SMPS measurements, lending support to assumption (a) (Isaacman-VanWertz et al., 2017; Isaacman-VanWertz et al., 2018). Assumption (b) may introduce bigger errors, particularly if sensitivity to compound $i$ is far from the

average, though we argue these errors are generally smaller for compounds that desorb at higher temperatures, as these are more likely to be larger molecules that contain multiple carboxyl or hydroxyl groups, both of which tend to reduce sensitivity variations (Lee et al., 2014).

### 3.6 Model implementation

The core of the model consists of a set of coupled differential equations, plus ancillary calculations, which are solved using MATLAB's *ode15s* solver. In the simple case of simulating evaporation of a single compound, but including the oligomerization terms (Eq. (11)), these are 8 differential equations, expressing the time derivatives of $T$, $C^*$, $k_g$, $k_d$, $N_g$, $N$, $k_{w,off}$ and $N_W$ (Eqs. (S3) to (S10)). The number of equations increases by extension to more than a single compound and by various options, such as deactivation of certain simplifications. The supplemental material contains details regarding the possible numbers of differential equations to be solved, and on the order of their evaluation in the solver.

### 3.7 Computational costs

A typical FIGAERO desorption experiment, as used here, lasts about 70 min: 20 min of ramping temperature up to 200 °C, followed by a 50-min "soak period" at a constant 200 °C. A single model run over one such desorption, for one or two compounds, takes less than a second on a mid-2010s 3-GHz MacBook Pro. However, an assumption of non-ideal heating was needed to explain observed tails in thermograms, at least for the calibration experiments described above (Fig. 3D). Its implementation currently consists of simply running the model several (e.g. 15) times, each time with a less efficient temperature ramp rate, and then calculating a weighted sum of the results, as illustrated in Fig. 4. With that, a model run takes several seconds to complete. Model performance takes further hits for each additional compound added to the model run, as the number of differential equations increases linearly with the number of modeled compounds (quadratically if using Eqs. (10a) and (10b)).

We typically run the model using only a single initial particle diameter $D_{P,0}$, as opposed to a size distribution, for the sake of reducing computational costs. In practice, the model results obtained from using the mass median diameter are very close to those obtained from using the actual size distribution, at least for the chamber experiments investigated here. Furthermore, as discussed below, the effect of particle size is lessened by the vapor-surface interactions that are assumed to occur subsequent to particle desorption.

Parameter optimization, i.e. finding the values for the free parameters that reproduce an observed thermogram, is currently still manual, requiring multiple model runs. The number of required runs depends on thermogram complexity and operator experience, 20 to 40 runs being typical. Future steps for making model application more efficient will be automation of that process through optimization algorithms, e.g. genetic algorithms.

## 4 General Model Behavior

### 4.1 Model sensitivity to volatility ($C^*$) and the $T_{max}$-$C^*$ relationship

Figure 5 illustrates the important role that vapor-surface interactions after desorption from the particle play in our model. The model explicitly calculates how many molecules of compound $i$ remain in the evaporating aerosol particle as a function of time, either in its free state ($N_i$) or low-volatility ("oligomer") state ($N_{i,g}$). For the simple case of a single-composition monodisperse aerosol, the time series the model obtains for $N_i$ shows a clear dependence on the compound's saturation vapor concentration ($C^*_0$): for a lower $C^*_0$, the particle evaporates within a higher temperature range, as expected (e.g. Fig. 5A). As described above, the model allows evaporating molecules to interact with (stick to) surfaces before entering the CIMS at a rate dependent on $C^*_0$ and the vapor-surface interaction parameters (Figs. 5B and C, respectively). The peak of recorded ion count rates in temperature space ($T_{max}$) shifts by about 15 to 20 °C for each order of magnitude of change in $C^*_0$.

Previous approaches in retrieving information from FIGAERO thermogram data have established that the measured $T_{max}$ values related roughly linearly to the logarithm of the saturation vapor pressures ($\sim C^*$), as shown for a set of well-characterized carboxylic acids (Lopez-Hilfiker et al., 2014). This $T_{max}$-$C^*$ relationship was subsequently used in more recent studies (D'Ambro et al., 2017; Mohr et al., 2017; Huang et al., 2018), and we used a subset of those calibration experiments here as initial verification of our model (Fig. 3), in particular the implementation of vapor-surface interactions, and for tuning our assumption of filter deposits not being heated equally efficiently (Fig. 4). Our model behavior reproduces the $T_{max}$-$C^*$ relationship in general (e.g., Fig. 5, Table 1). In the following sections, we will see how other model input parameters affect $T_{max}$ as well, and revisit in section 4.4. the model reproduction of the $T_{max}$-$C^*$ relationship.

### 4.2 Effect of vapor-surface interactions

If vapor-surface interactions were ignored, all $T_{max}$ would be shifted lower, viz. to the temperature where the steepest decrease in $N_i$ occurs in Fig. 5A. Table 1 presents model-obtained $T_{max}$ values for the same range of $C^*_0$, from 1 to $10^{-6}$ µg m$^{-3}$, and also for a range of $D_{P,0}$, from 5 to 500 nm, both for the default case of vapor-surface interactions implemented and for the case of ignoring these interactions. As expected, the difference in calculated $T_{max}$ between these cases is most pronounced for smaller particles (fast particle evaporation) and lower volatilities (long subsequent residence time on surfaces). However, the signal obtained by FIGAERO particle desorption measurements is proportional to deposited mass, and sufficient mass is required for acceptable signal-to-noise ratios. Therefore, the majority of measurements by FIGAERO are made on aerosol with mass median diameters >100 nm. Consequently, negligence of the vapor-surface interactions when applying the model to observations would be compensated by under-estimating $C^*_0$ by typically an order of magnitude, as shown through Table 1.

**4.3 Limitations to evaporations described by $\alpha < 1$**

Figure 6 presents changes in model output, in terms of normalized thermograms, when certain input parameters are varied individually for equally simple model runs. In most cases, we do not expect to be able to distinguish a lower $C_0^*$ from a lower evaporation coefficient $\alpha$, the latter for instance a result of potential in-particle diffusion limitations (cf. Figs. 6A & 6B). That is, of course, provided we do not have prior knowledge of either input parameter. Variations of relatively high values of $\alpha$ may also go entirely unnoticed, when the eventually recorded signal is controlled by the post-evaporation vapor-surface interactions, which do not anymore depend on $\alpha$ as it is specified for evaporation from the particles. However, this masking effect is a smaller issue for (more relevant) larger particles.

Ambiguities between possible diffusion limitations ($\alpha < 1$) versus merely a lower $C_0^*$ could be addressed by future blank experiments, i.e. with a particle filter in the inlet line to prevent deposition of particles on the FIGAERO filter, in particular when also implementing periods of isothermal evaporation of various durations. We actually used such an experiment here as a rough confirmation of our model implementation of vapor-surface interactions (Fig. 2). An obvious advantage of blank experiments is that any effects due to evaporation from particles are removed, i.e. $\alpha = 1$. However, the measurements are restricted to such gas-phase compounds that deposit on the filter (and other surfaces) despite the blanking filter. Therefore, only semi-volatile compounds are detectable, and in particular for larger typical terpene oxidation products it may not be guaranteed that the observed compositions actually correspond to the same isomers observed during particle desorption. In addition, compounds could arise from the decomposition of low- or non-volatile compounds that have remained on the filter from preceding experiments, an issue that also emphasizes the importance of using sufficiently clean filters. For these reasons, blank experiments may be more useful for chemically simple systems. An additional caveat is that sampled gas-phase compounds may deposit on more surfaces than aerosol particles do.

We defer this type of investigation to future, dedicated experimental studies, and will simply assume $\alpha = 1$ in most of the remainder of this study.

**4.4 Model sensitivity to other input parameters**

As the vaporization enthalpy $\Delta H$ controls the increase $C^*$ with increasing temperature (Eq. (2)), it is a very powerful handle on the thermogram shape and the main factor determining the initial upslope of the thermogram as well as peak width (Fig. 6C). Varying the vapor-surface interaction parameter $C_W$ (Fig. 6D) shifts the thermograms as expected from the discussion above (cf. Fig. 3). Resulting from these interactions, an additional masking effect becomes apparent when comparing results from varying the initial particle diameter $D_{P,0}$ (Fig. 6E). In our case here, practically no effect is expected for variations of $D_{P,0}$ below 500 nm (see also Table 1). Lastly, the temperature ramp rate merely shifts the modeled thermograms towards higher temperatures for faster ramps (Fig. 6F). However, the shift is typically small, i.e. less than 10 °C for a change in ramp rate by a factor of two.

Figure 7 summarizes how, for otherwise typical assumptions and conditions, the simulated $T_{max}$ is defined by $C^*_0$ and $\Delta H$. (colored line), generalizing the $T_{max}$-$C^*$ relationship found previously based on experimental observations (Lopez-Hilfiker et al., 2014; Mohr et al., 2017; colored circles and black line). As the colors of the circles roughly match those of the underlying model-derived lines, the model largely reproduces the empirical relationship, as seen above (section 4.1, Fig. 3D). Conversely, comparison with the model results infers a relation between $C^*_0$ and $\Delta H$ (colored lines vs. black line), which consistently predicts relatively lower values for $\Delta H$ than an independent semi-empirical $C^*_0$-$\Delta H$ relation (Epstein et al., 2010; black dashed line).

## 4.5 Inclusion of oligomer formation and dissociation

When oligomer formation and dissociation reactions are included, model runs are initiated with the molecules of compound $i$ distributed between a high-volatility state (monomer) and a low-volatility state (e.g. in oligomer), at the fraction resulting from assuming equilibrium between the reactions (section 3.4). Four additional free parameters control these reactions (Eqs. (12) and (13)) and offer a large amount of conceivable combinations of values. In practice, the model is now able to obtain a substantially increased variety of thermogram shapes (Fig. 8). In particular, the typical peak can be extended towards higher desorption temperatures by adding/accentuating features such as tails, shoulders or secondary peaks. In total, it appears a wise choice of model parameters would obtain many relevant (i.e. observed) thermogram shapes, at least in reasonable approximation (cf., e.g., Lopez-Hilfiker et al., 2015; Lopez-Hilfiker et al., 2016b; D'Ambro et al., 2017; Huang et al., 2018). In practical fitting to experimental data, it was typically best to first obtain an approximate value for $k_{d,0}$, the oligomer dissociation rate at room temperature, which greatly affects the shape of the tail (Fig. 8C). The activation energy $E_d$ affects how quickly $k_d$ increases in the temperature ramp. A relatively low value (e.g. $E_d < 20$ kJ mol$^{-1}$) makes for a fairly flat tail, whereas high values can lead to a shoulder or secondary peak, with its $T_{max}$ moving towards lower temperatures as $E_d$ is increased when all other parameters remain unchanged (Fig. 8A). In reality however, we expect a higher $E_d$ to be coupled to a lower $k_{d,0}$, as the pre-exponential factor in the actual Arrhenius relation is typically a constant,

$$k_d = A \cdot e^{-\frac{E_d}{RT}} \tag{17}$$

thus coupling $E_d$ and $k_{d,0}$:

$$k_{d,0} = A \cdot e^{-\frac{E_d}{RT_0}} \tag{18}$$

Below, we revisit this expected relationship in modeling thermogram data of chamber-generated SOA, but in general $E_d$ and $k_{d,0}$ remained independent model parameters.

The main role of the oligomer formation rate at room temperature $k_{g,0}$, in practice, was to control the relative amount of compound $i$ that is present in the non-volatile (oligomer) state at the beginning of the desorption, at room temperature (Fig. 8D). As described above, that fraction is determined by $k_{g,0}/(k_{g,0}+k_{d,0})$, i.e. we are assuming steady-state conditions in the collected aerosol initially. The corresponding activation energy, $E_g$, turned out to have only a small effect on the modelled thermograms (Fig. 8B), especially for relatively small values. This is because as the initial steady-state is perturbed, net

oligomer dissociation occurs right from the beginning of the desorption. Furthermore, $E_g$ is expected to be relatively small value, in particular smaller than $E_d$, if a major fraction of the measurements is to be explained by oligomer dissociations. Accordingly, previous studies employing the same conceptual approach have used $\Delta E = E_d - E_g = 15\text{-}42$ kJ mol$^{-1}$ (Kolesar et al., 2015a) or $E_g = 0$ (Trump and Donahue, 2014). Low values for $E_g$ may also be supported by studies on carboxylic acid

dimers, which have shown them to form (via hydrogen exchange) with activation energies $< 6$ kJ mol$^{-1}$ (e.g., Meier et al., 1982; Loerting and Liedl, 1998). For the remainder of this study, we always used $E_g = 0$.

As a consequence of the small practical role of the oligomer formation parameters in shaping the modeled thermogram, the desorption modeling of compound $i$ does not directly distinguish between thermal decomposition of reversible oligomers and thermal decomposition of other low volatility compounds, e.g. parent compounds formed independently of monomer $i$ or

parent compounds formed in the gas-phase. Rather, the assumption of reversible oligomers when applying the model to produce observed thermograms infers a corresponding formation rate constant ($k_{g,0}$), based on the required initial in-particle partitioning between free monomers ($N_{i,0}$) and those to be sourced from thermal decomposition processes ($N_{i,g,0}$). In any case, model application in this way can provide insights into the possible bond dissociation energies that produce the observed thermograms.

**5 Model Application to Chamber-Generated SOA**

**5.1 Modeling, e.g., the C$_8$H$_{12}$O$_5$ thermogram: three illustrative approaches**

Applying the model to observations of actual SOA allows characterization of the SOA in terms of the effective volatility of individual detected compositions. We used experimental data obtained from monoterpene-derived SOA generated in a continuous flow reaction chamber at PNNL (section 2). Particles were sampled from the chamber when steady state

conditions prevailed. Figures 9-11 illustrate model results and comparisons with experimental thermogram data for the composition C$_8$H$_{12}$O$_5$. Model parameters used are summarized in Table 2. The data were taken during conditions of dark ozonolysis of α-pinene at concentrations of ozone (O$_3$) at 84 ppbv and of α-pinene reacted at 6.7 ppbv. C$_8$H$_{12}$O$_5$ was one of the dominant compositions observed in the particle-phase by FIGAERO-iodide-CIMS for this system. In addition, the thermogram characteristics for C$_8$H$_{12}$O$_5$ were typical for many compositions for our experiments at PNNL, i.e. a main peak

followed by a shoulder and an exponential decay of desorption during the soak period. Our measured thermogram shapes for a given chamber condition were highly reproducible (supplemental material, Fig. S1), as expected from previous studies (Lopez-Hilfiker et al., 2014), and we therefore neglect experimental uncertainties in the following. However, we generally do expect changes in thermogram shapes for individual compositions, if there are changes in the instrumental setup (section 5.6) or experimental conditions.

The SOA formed in the chamber was not monodisperse. For the experiment discussed here, the size mode diameter was $<$ 100 nm, while the volume median diameter was 197 nm (geometric standard deviation of the volume size distribution = 1.5). This volume median diameter was used for the model predictions shown in Figs. 9-12, because the resulting model output

was nearly indistinguishable from the results obtained using the actual number size distribution, while the model calculations were much faster. The experiments were seeded with 50 nm ammonium sulfate particles, so we also included a refractory (non-volatile) core of that diameter to the modeled aerosol (see supplemental material). The calculated filter loading for this experiment was 0.31 µg. No matrix effects were apparent.

### 5 5.1.1 Approach 1: no thermal decomposition

For the first approach (Fig. 9), the model was run without the oligomerization terms, but instead assumed that the $C_8H_{12}O_5$ thermogram was the result of five compounds desorbing independently, each as per Eq. (1). The hypothetical compounds differ from each other by factors of 10 in their saturation concentration at room temperature (298.15 K). This approach is analogous to assuming that there are five different structural isomers of $C_8H_{12}O_5$ having different functional groups and thus

volatilities. Whether this is reasonable in this specific case is irrelevant for our purposes here. We assumed $\alpha = 1$, and for the most volatile compound, we constrained $\Delta H$ by the initial thermogram slope ($\Delta H = 105$ kJ mol$^{-1}$). For simplicity, this $\Delta H$ was used for all compounds as well, an assumption addressed below.

The $C_8H_{12}O_5$ thermogram is successfully produced by the model using this 5-bin volatility basis set (VBS) when inferring relative bin-wise contributions of 10% to 40% (Fig. 9B). However, this approach fails in reproducing the exponential decay

of desorption signal during the soak period (i.e. at constant 200 °C), which in the simulation proceeds much faster than observed (Fig. 9C). A remedy would be extending the VBS towards more bins at yet lower volatility, but with high relative contributions, namely at the level of the highest-volatility bin. However, it seems unlikely that a single elemental composition, such as $C_8H_{12}O_5$, would have dozens of structures ranging over five (or more) orders of magnitude in volatility. It is even less likely that such a VBS would predict the soak period correctly while also maintaining a flat thermogram

shoulder. As noted, we used the same $\Delta H$ for all hypothetical compounds representing $C_8H_{12}O_5$, although $\Delta H$ is generally expected to increase with lower volatility. Epstein et al. (2010) suggested an increase of $\Delta H$ by 11 kJ mol$^{-1}$ for each decade of lower saturation concentration. Such a relationship between $C^*_0$ and $\Delta H$ would both narrow the peaks of the five individual compounds and lower their $T_{max}$, (Fig. 9A, cf. Fig. 6), and thus require an even greater (and less likely) range of volatilities to predict the observed thermogram.

Lopez-Hilfiker et al. (2014) previously fitted thermograms of individual compositions of α-pinene SOA simply by using a variable number of peaks of a certain peakshape. The peakshape was first obtained from presumably single-component thermograms, and the fitting procedure allowed the location (in temperature space), amplitude and width of the peaks to vary. That is a rather heuristic approach, but allowed for comparison with the $T_{max}$-$C^*$ relationship established in the same study, leading to a similar result as our Approach 1. Correspondingly, they reached the same conclusion, as above, that the

much lower volatilities inferred by desorption at temperatures much higher than the observed primary peak are inconsistent with the observed compositions and conceivable isomers.

**5.1.2 Approach 2: only thermal decomposition, and $E_d$ coupled with $k_{d,0}$ (Arrhenius)**

For the second approach (Fig. 10), we explored the opposite extreme: Could the thermogram be explained *only* by thermal decomposition of some non-volatile material into $C_8H_{12}O_5$. We set $N_{i,0} = 0$, and for simplicity excluded from the model a possible reverse reaction, i.e., formation of nonvolatile material by $C_8H_{12}O_5$. Formally, this approach corresponds to using

Eq. (11) without the last term. Volatility parameters for $C_8H_{12}O_5$ monomers were chosen sufficiently high ($C^*_0 = 2$ µg m$^{-3}$, $\Delta H = 80$ kJ mol$^{-1}$), so that the decomposition into $C_8H_{12}O_5$ would itself be the process limiting the desorption rate. Rather than freely fitting a room temperature decomposition rate $k_{d,0}$ and activation energy $E_d$ (Eq. (12)), we required the rate and activation energy to be related as per the actual Arrhenius relation (Eq. (18)). For the pre-exponential factor $A$, we assumed 3 $\times$ 10$^{10}$ s$^{-1}$, as previously used for bimolecular dissociation reactions (Trump and Donahue, 2014).

With these conditions set, we needed a range of bonds with different activation energies undergoing decomposition in order to generate a wide shoulder in the thermogram, as opposed to a relatively narrow secondary peak. When allowing an arbitrary but reasonable range of activation energies, from 80 to 115 kJ mol$^{-1}$ (Fig. 10B), we could indeed reproduce the $C_8H_{12}O_5$ thermogram well with the model under the assumptions of this approach, namely that *all* of it arises from thermal decomposition (Fig. 10). The sensitivity of these results to the pre-exponential factor $A$ is moderate because variation of $A$ by

one order of magnitude can be compensated by adjusting activation energies by roughly $\pm 10$ kJ mol$^{-1}$.

Analogously to Approach 1 above, the model underestimates the observations during the soak period (Fig. 10C), which may be expected given the similarity of Eq. (17) to Eq. (2), which controlled evaporation rates at 200 °C under Approach 1.

**5.1.3 Approach 3: thermal decomposition of reversible oligomers, and $E_d$ as free parameter**

For the third approach (Fig. 11), we included both oligomerization terms of Eq. (11), i.e. oligomer dissociation as well as

formation, with the initial conditions obtained via the assumption of steady state between monomers and oligomers, as described above. Unlike Approach 2 (Eqs. (17) and (18)), the rate constants and respective activation energies are independent free parameters (Eqs. (12) and (13)). We characterize the monomer by a single volatility: $C^*_0 = 0.8$ µg m$^{-3}$ (chiefly determining $T_{max}$), $\Delta H = 88$ kJ mol$^{-1}$ (as required by the initial slope), and $\alpha = 1$ (arbitrarily set). By choosing the parameters controlling oligomerization $k_{g,0} = 1.7 \times 10^{-3}$ s$^{-1}$, $k_{d,0} = 5 \times 10^{-4}$ s$^{-1}$, $E_g = 0$ kJ mol$^{-1}$ and $E_d = 6$ kJ mol$^{-1}$, the model

could reproduce the entire thermogram well, now also including the soak period (Fig. 11C). As shown through Fig. 11D, the molecules initially free to evaporate, i.e. present as high-volatility monomers, evaporate quickly, giving rise to the main peak of the thermogram. These parameters predict 77% of $C_8H_{12}O_5$ is initially in a non-volatile state, of which the relatively slow decomposition into $C_8H_{12}O_5$ (and other compositions) accounts for the high-temperature "shoulder" of the observed thermogram. The simulation is not particularly sensitive to $E_g$, as any sufficiently low value ($E_g < 15$ kJ mol$^{-1}$) will not lead

to appreciable oligomer formation during heating. Therefore, setting $E_g = 0$ appears to be the simplest choice (see also section 4.5).

In summary, this approach performs especially well in reproducing the observations, by two criteria: a) it requires the least number of free parameters compared to the other approaches, and b) it is straight forward to also explain soak periods with relatively shallow decay of signal. The other two approaches could in principle reproduce these soak periods as well, but that would require either a very specifically tuned VBS (Approach 1) or activation energy distribution (Approach 2).

However, it is important to note that this success of Approach 3 is primarily founded on the independence of $E_d$ from $k_{d,0}$, and secondarily on allowing a mix of free monomers and monomers sourced from decomposition (here at a 23:77 ratio), and not necessarily on the reversible character of the low-volatility state. The pre-exponential factor for dissociation resulting from $k_{d,0}$ ($5 \times 10^{-4}$ s$^{-1}$) and $E_d$ (6 kJ mol$^{-1}$) is $A_d = 5.6 \times 10^{-3}$ s$^{-1}$ (Eqs. (12) and (18)), much lower than the value of $A = 3 \times 10^{10}$ s$^{-1}$ presumed in Approach 2. And when using these $k_{d,0}$ and $E_d$, together with the initial condition that decomposition
accounts for 77% of $C_8H_{12}O_5$, we can even omit the oligomer formation term in Eq. (11) in the model and obtain practically the same thermogram. This similarity is because in practice, no new reversible oligomers would actually form upon heating (Fig. 11D). As discussed above (section 4.5), model application under the assumption of initial steady-state between monomers and reversible oligomers delivers a rate constant for formation of these oligomers, $k_{g,0}$, here $1.7 \times 10^{-3}$ s$^{-1}$. The assumption therefore infers that $C_8H_{12}O_5$ enters oligomers (to re-iterate, "oligomers" meaning a not more closely defined non-volatile state) at a time scale of 10 minutes.

## 5.2 Model application to thermograms of other SOA compositions

A large number of compositions detected from dark α-pinene ozonolysis produced thermograms qualitatively similar to that for $C_8H_{12}O_5$, i.e. featuring a prominent peak, followed by a fairly flat shoulder towards higher temperatures. These thermograms can be modeled in the manners discussed above. However, there is a great variety of thermogram shapes overall, and a substantial number of compositions exhibit qualitatively different thermograms. Four such example cases are
presented in Figs. 12 and 13, with model input parameters provided in Table 3.

Figure 12A shows how the thermogram for $C_8H_{10}O_5$ can be modeled well, in the same way that produced the best results for $C_8H_{12}O_5$ (Fig. 11; i.e. Approach 3), i.e. by assuming a single compound evaporating but assuming a major fraction stems from the decomposition of reversibly formed oligomers. For $C_8H_{10}O_5$ however, the thermogram is more clearly bimodal. These features in turn require appropriate adjustments to the model input parameters, most notably a higher $E_d$ at 17 kJ mol$^{-1}$
(vs. 6 kJ mol$^{-1}$ for $C_8H_{12}O_5$). Although $T_{max}$ is practically the same for either composition (83 and 84 °C), the volatility used for $C_8H_{10}O_5$ is higher ($C^*_0 = 4$ μg m$^{-3}$, vs. 0.8 μg m$^{-3}$ for $C_8H_{12}O_5$) to compensate for the lower vaporization enthalpy ($\Delta H$) required by the shallower initial slope.

In general, the routine as per Approach 3 can be used to explain two main features of a given thermogram, such as two
separate peaks, or one peak and a shoulder. The soak period can be typically fit at the same time. The thermogram for $C_{10}H_{14}O_5$, however, has a higher $T_{max}$ (100 °C), a typical high-temperature shoulder, and also exhibits a shoulder at 79 °C (Fig. 12B). All three of these features can be reproduced by assuming two compounds desorbing independently. For reproducing the shoulder, the less volatile compound is modeled as per Approach 3, i.e. including oligomer dissociation

(hence shown in blue). Its relatively lower saturation concentration of $C^*_0 = 0.08$ µg m$^{-3}$ reflects the high $T_{max}$. Inclusion of a more volatile compound ($C^*_0 = 0.3$ µg m$^{-3}$) facilitates the low-temperature shoulder (shown in beige).

The compositions we have considered this far are products of α-pinene oxidation that have retained most of their carbon backbone. We would expect that the composition $C_2H_2O_3$, for instance, behaved quite differently. This specific composition is observed here as one of the more common organics that have lost most of their carbon, presumably either through fragmentation during oxidative processing (in gas- and/or particle-phase) prior to heating, or through thermal decomposition during heating. Interestingly, the $C_2H_2O_3$ thermogram does not, at first glance, appear very different from the thermograms obtained for larger compositions. It features a double peak at 90-125 °C, followed by a shoulder (Fig. 12C). However, the signal decay during the 200-°C soak period is strikingly steeper compared to the observations for $C_8H_{10}O_5$, $C_8H_{12}O_5$ or $C_{10}H_{14}O_5$. Indeed, the decay is consistent with our model simulation of thermal decomposition as explored in Approach 2 above (section 5.1.2), i.e. following the Arrhenius relation (Eq. (17)) with $A = 3 \times 10^{10}$ s$^{-1}$, specifically with an activation energy of $E_d = 111$ kJ mol$^{-1}$. We therefore chose to model the $C_2H_2O_3$ thermogram as arising purely from evaporation that is limited by the thermal decomposition of parent compounds at a chosen distribution of activation energies (Table 3), i.e. in essentially the same manner as done for Fig. 10. The respective activation energies ranged from 76 to 121 kJ mol$^{-1}$, with the median of the distribution at 101 kJ mol$^{-1}$. The parent compounds are obviously required to decompose prior to desorbing themselves. We used the model to estimate the corresponding upper-limit volatilities, which ranged from $C^*_0 = 8$ µg m$^{-3}$ to 5 $\times 10^{-9}$ µg m$^{-3}$, with the median activation energy requiring $C^*_0 < 2 \times 10^{-4}$ µg m$^{-3}$. (This estimate implicitly also assumes that all of the parent compounds produce $C_2H_2O_3$ as they decompose.)

Finally, we take a closer look at the thermogram for $C_{18}H_{28}O_6$, one of the dimers of α-pinene oxidation products observed at acceptable signal-to-noise for this experiment, and one relatively little affected by other ions detected at the same integer mass-to-charge ratio. As expected for a much larger molecule, the $C_{18}H_{28}O_6$ thermogram's $T_{max}$ (121 °C) is higher than in previously considered cases. Unlike the previously considered thermograms, the $C_{18}H_{28}O_6$ thermogram could be explained by simple single-compound evaporation, i.e. without invoking decomposition processes (Fig. 13A). For Fig. 13A however, we used $C^*_0 = 0.25$ µg m$^{-3}$, a seemingly high value for a $T_{max}$ of 121 °C, for example when compared to the high-volatility $C_{10}H_{14}O_5$ component ($C^*_0 = 0.3$ µg m$^{-3}$), which peaks at 75 °C. Until this point, despite having typically involved decomposition processes, the observed $T_{max}$ and the corresponding model parameters (Figs. 9-12, Tables 2 and 3) have roughly agreed with the $T_{max}$-$C^*$ relationship as established by Lopez-Hilfiker et al. (2014). But 0.25 µg m$^{-3}$ lies considerably above the value of ~ $10^{-5}$ µg m$^{-3}$ suggested by that relationship when extrapolated to $T_{max} = 121$ °C (from data in the range 35 to 90 °C; Mohr et al., 2017). This discrepancy is explained partly by the small Raoult factor for $C_{18}H_{28}O_6$, as derived from relative signal intensities (Table 3), but chiefly by the surprisingly low $\Delta H$ that the model suggested due to the shallow slope of the low-temperature side of the thermogram peak (60 kJ mol$^{-1}$ for $C_{18}H_{28}O_6$, compared to 115 kJ mol$^{-1}$ for $C_{10}H_{14}O_5$). If we forewent fitting the thermogram's upslope and, for instance, used $\Delta H = 120$ kJ mol$^{-1}$, we would get $T_{max} = 121$ °C by decreasing $C^*$ to ~ $10^{-3}$ µg m$^{-3}$. In conclusion, the low $\Delta H$ used for modeling the $C_{18}H_{28}O_6$ thermogram in Fig. 13A is likely

unphysical and strongly suggests that the situation is more complicated, i.e. the desorption signal from $C_{18}H_{28}O_6$ is not simply due to direct evaporation of a single compound.

An alternative way to modeling the $C_{18}H_{28}O_6$ thermogram is assuming it is itself the product of thermal decomposition. Using the simple strategy presented above (Approach 2, $A_d = 3 \times 10^{10}$ s$^{-1}$), the model indeed obtained a good fit using a single activation energy, $E_d = 92$ kJ mol$^{-1}$ (Fig. 13B). The consequent dissociation rate constant at room temperature is $k_{d,0} = 2.8 \times 10^{-6}$ s$^{-1}$, corresponding to a lifetime of 100 hours. Although we did not observe dimers in the gas-phase in our experiments, that may be simply due to efficient partitioning into the particle phase, as previous studies have suggested that dimers of α-pinene oxidation products do form in the gas-phase (Ehn et al., 2014; Mohr et al., 2017). Even so, they may react into a yet lower-volatility state in the particle phase. However, we suggest another alternative solution, namely that the thermogram signal is the superposition of the direct evaporation of three (or more) $C_{18}H_{28}O_6$ isomers, with their respective volatilities ranging from $C^*_0 = 2 \times 10^{-2}$ to $5 \times 10^{-5}$ µg m$^{-3}$ (together with $\Delta H = 120$ to 146 kJ mol$^{-1}$, Table 3, Fig. 13C), a conceivable range for a large multi-functional elemental formula, and in particular also in the range we expect (cf. Mohr et al., 2017).

### 5.3 Evaporation coefficients ($\alpha$)

We want to re-iterate here that we used an evaporation coefficient of $\alpha = 1$ throughout section 5. But as expected from discussion above (section 4.3), equally good fits could generally be obtained under the assumption of a lower $\alpha$, by adjusting the free parameters accordingly, in particular $C^*_0$ and $\Delta H$. For example in section 5.1.3 (Approach 3), a value of $\alpha = 0.1$ would result in $C^*_0 = 5$ µg m$^{-3}$, $\Delta H = 65$ kJ mol$^{-1}$ and $k_{g,0} = 1.3 \times 10^{-3}$ s$^{-1}$ (vs. $C^*_0 = 0.8$ µg m$^{-3}$, $\Delta H = 88$ kJ mol$^{-1}$ and $k_{g,0} = 2 \times 10^{-3}$ s$^{-1}$ when $\alpha = 1$). Evaporation coefficients between 0.1 and 0.2 were suggested e.g. by Saleh et al. (2013) for SOA derived from α-pinene (in somewhat different conditions, e.g. relative humidity < 10%). We also note here though that α much lower than 0.1 in turn require unrealistically high values of $C^*_0$ and $\Delta H$ considering the elemental compositions (e.g., $C_8H_{12}O_5$).

### 5.4 Timescales of oligomeric material's dissociation (and formation, where applicable)

Our implementation of principally reversible oligomerization, as per Eq. (12), is based on previous, generally successful attempts of describing SOA particle growth and/or evaporation upon dilution or heating through inclusion of such processes (Trump and Donahue, 2014; Kolesar et al., 2015a). And in several points, our results quantitatively agree with those studies. Trump and Donahue (ACP, 2014), for instance, achieved good agreement between model and experiment when the dissociation of oligomers was effectively controlling SOA evaporation upon dilution. Based on observed evaporation time scales on the order of an hour, they used $k_{d,0} = 10^{-4}$ s$^{-1}$, together with formation rates of $k_{g,0} = 10^{-2}$ to $10^{-1}$ s$^{-1}$ that were consistent with observations of SOA growth. When implementing reversible oligomerization in a more complex model of aerosol chemistry and dynamics, values in the same ranges achieved best overall agreement with observations: $k_{d,0} = 10^{-5}$ to $3 \times 10^{-3}$ s$^{-1}$ and $k_{g,0} = 10^{-2}$ s$^{-1}$ to $10^{-1}$ s$^{-1}$ (Roldin et al., 2014). These numbers are in good agreement with the values we have

derived here (Tables 2 and 3). Also Kolesar et al. (2015a) employed essentially the same reversible oligomerization equations for fitting a model to SOA heating-induced desorption and isothermal evaporation data from Vaden et al. (2011), obtaining somewhat higher values, corresponding to the ranges $k_{d,0} = 10^{-3}$ to $3 \times 10^{-2}$ s$^{-1}$, and $k_{g,0} = 2$ to $\sim 10^5$ s$^{-1}$.

All in all, the dissociation rates we obtain from applying our model to FIGAERO thermogram data are mostly consistent with previously derived values. They correspond to room-temperature lifetimes ($1/k_{d,0}$) of 20 to 90 minutes when using low activation energies ($E_d < 25$ kJ mol$^{-1}$), in model simulations consistent with reversible oligomerization (Figs. 11, 12A, 12B). When initial steady-state of reversible oligomerization is assumed, the formation time scales ($1/k_{g,0}$) are 1 to 15 minutes. When using high activation energies ($E_d > 80$ kJ mol$^{-1}$; Figs. 12C, 13B), the inferred lifetimes against dissociation at room temperature are much longer and wide-ranging: an hour to years.

## 5.5 Activation energies $E_d$ as indicator for the type of thermal decomposition processes

Despite the ambiguities remaining in applying our model to FIGAERO data, we do obtain from it stricter constraints on dissociation kinetics, allowing more qualified conclusions on the underlying decomposition mechanisms. Various detailed mechanisms have previously been proposed (e.g., Lopez-Hilfiker et al., 2015):

(a) Thermal decomposition through O–O bond cleavage. The O–O bond is in general the weakest covalent bond in oxygenated organic molecules. For organic peroxides, the O–O bond strength is traditionally $\sim 140$ kJ mol$^{-1}$, but may cover a range from $\sim 90$ to 200 kJ mol$^{-1}$ (Bach et al., 1996). For some peroxyhemiacetals, activation energies for O–O bond cleavage have been reported considerably lower, between 33 and 46 kJ mol$^{-1}$ (Antonovskii and Terent'ev, 1967).

(b) Cleavage of other covalent bonds. For instance, certain carboxyl and hydroxyl groups can undergo heat-induced dehydration and decarboxylation already at temperatures of 200 °C (Canagaratna et al., 2015). For citric acid ($C_6H_8O_7$), such fragmentation has been directly shown to occur during FIGAERO thermal desorption experiments, starting at < 150 °C (Stark et al., 2017).

(c) Thermal break-up of a matrix of non-covalent H bonding. In total, this matrix may be stronger in SOA than in ideal mixtures or pure liquids or solids. Individual H bonds between organic molecules can be very weak, with activation energies computed as low as $\sim 2$ kJ mol$^{-1}$, i.e. extending into the regime of van der Waals-type interactions, and range up to $\sim 30$ kJ mol$^{-1}$ for interactions between carboxyl moieties (Steiner, 2002).

Mechanisms (a) and (b) could both be a source and a sink of high-temperature signal for a certain composition during a desorption run; a source via decomposition of any parent compounds, a sink via decomposition of the considered composition itself. Mechanism (c) is more likely to appear as a source than a sink, as even the smallest matrix precursor, a two-molecule cluster, would be of relatively low volatility and therefore likely dissociate before it would desorb upon heating. In addition, organic molecular clusters bound by H-bridges almost certainly break inside the CIMS. Literature suggests that bindings with enthalpies smaller than roughly 100 kJ mol$^{-1}$ have a high probability of break-up in the instrument's atmospheric pressure interface (Iyer et al., 2016; Lopez-Hilfiker et al., 2016a; Frege et al., 2018), practically eliminating the chances of detecting such hypothesized parent structures directly.

Applying our model to SOA from α-pinene ozonolysis, we achieve remarkably good fits to thermograms of many major compositions when assuming that relatively low-volatility material is dissociating with low activation energies ($E_d < 25$ kJ mol$^{-1}$). Because we expect $E_g$ to be very small and used $E_g = 0$ here (section 4.5), we are in rough agreement in terms of $\Delta E = E_d - E_g$ with the range given by Kolesar et al. (2015a), ($\Delta E = 15$ to 42 kJ mol$^{-1}$, with smaller values more likely). If we accept that these low activation energies explain the high-temperature behavior of many individual (monomers') thermograms, the dissociating oligomers may not actually be covalently bound monomers. Matrices established by a network of weak H bonds, which substantially decrease the components' volatility compared to ideal mixtures, is the mechanism most consistent with our combined observations and modeling (mechanism (c)). Despite low activation energies, the dissociation rates at room temperature ($k_{d,0}$) are small, on the order of 1 h$^{-1}$. In terms of the Arrhenius relation (Eq. (17)), these energies thus infer very slow pre-exponential factor $A$, from ~ 10 s$^{-1}$ down to ~ 10 h$^{-1}$, which may be indicative of a more complex decomposition mechanism than heat-induced cleavage of simple bonds.

The high-temperature part of thermograms can in principle also be explained by thermal decomposition at higher activation energies, i.e., $E_d > 50$ kJ mol$^{-1}$, in most cases however requiring not only one value, but a distribution of bonds with differing $E_d$ (Figs. 10, 12C). Especially for relatively small oxygenated organic compositions (e.g. $C_2H_2O_3$, Fig. 12C), the agreement between predicted and observed thermograms is convincing and suggests that we are able to attribute the full thermogram to the thermal decomposition of less volatile organic material. This result is compatible with the expectations that such small compositions would otherwise be too volatile to partition into the particle phase at all. Activation energies for decomposition of around 100 kJ mol$^{-1}$ suggest O–O bond cleavage as a source (Bach et al., 1996; mechanism (a)).

Previous FIGAERO measurements of SOA, likely in somewhat different conditions, have also produced relatively narrow secondary peaks at high desorption temperatures, standing mostly separate or superimposed on shoulders (e.g., Lopez-Hilfiker et al., 2015). These features were observed in individual thermograms, specifically shown for large (#C > 7) monoterpene oxidation products, and suggest a contribution of cleavages of covalent O–O bonds to the respective desorption signals.

## 5.6 Noted challenges

A potential issue, unaddressed in the current model, is that heat-induced decomposition can also be a sink of signal within an individual thermogram, in particular via mechanisms (a) and (b) discussed above. For citric acid, for instance, such decomposition leads to a faster-than-expected drop of the corresponding signal as the desorption temperature is ramped up, while compositions corresponding to expected decomposition products can be observed (Stark et al., 2017).

Stark et al. (2017) also pointed out that there can be large differences in the response of individual FIGAERO instruments during calibration experiments, in particular regarding $T_{max}$, and in particular when differences in the exact instrument designs are involved. These discrepancies imply the need of adjusting our model for application to data from different instruments. Such model calibration can likely be achieved as described in section 3.3. It appears likely that model re-

calibration is also necessary whenever the sampling or thermal desorption geometry of a specific instrument has changed, in particular the heater setup including the position of the thermocouples used for measuring the desorption temperature profile. Another issue, worth pointing out again, is the possible errors introduced if there are indeed multiple isomers contributing to a single composition's thermogram, if their volatilities ($C^*_0$) differ, but not by enough to be revealed by separate thermogram peaks. We show a possible ambiguity of this type for $C_{18}H_{28}O_6$ (cf. Fig. 13A and 12C; Table 2). The primary effect of simulating an observation of a single thermogram peak by assuming multiple isomers (i.e., multiple $C^*_0$), instead of a single isomer, is that overall lower $C^*_0$ and higher $\Delta H$ need to be used.

Finally, there are challenges related to the CIMS detector. Most notably for experiments on SOA, it is very common to measure more than one elemental composition at an integer (nominal) mass-to-charge ratio, requiring a sufficient resolving power for assigning the correct compositions (Stark et al., 2015; Cubison and Jimenez, 2015). An additional requirement here is that the mass axis calibration needs to be stable or maintained precisely throughout aerosol desorption, such that drifts in calibration do not occur or are accounted for. Such drifts would cause artificial shifts in how signal is distributed between compositions of same nominal mass, and substantially affect the shapes of the respective thermograms.

### 5.7 Additional constraints through including isothermal evaporation phases

For reducing ambiguities when applying our model, we recommend dedicated sets of experiments that go beyond the classical desorption schedule consisting of a linear temperature ramp plus soak period. The main goals are to reduce ambiguities regarding which processes best simulate aerosol evaporation, and to confirm or improve the implementation of vapor-surface interactions subsequent to aerosol desorption. In section 4.3 above, we already suggested some dedicated experiments aimed at retrieving constraints on possible diffusion limitations ($\alpha < 1$). Another recommendation are sets of desorption experiments on SOA particles that include periods of isothermal evaporation of a variety of lengths, in particular at room temperature, or the temperature at which the SOA was generated. Experiments similar to what we are suggesting here are the earlier isothermal dilution experiments using other detectors (Wilson et al., 2015). The straightforward approach here would be simply delaying the start of heating, while observing SOA particle evaporation from the filter upon mere removal of the gas-phase, followed by a standard temperature ramp to desorb the remaining material (D'Ambro et al., 2018). The chief advantage is that all temperature-dependent parameters are irrelevant for a period of isothermal evaporation, most importantly activation energies and vaporization enthalpies. At room temperature, any artifacts from imperfect heating (section 3.3) are removed as well. Also, such experiments can encompass many hours, allowing investigation of slow processes that become apparent only at larger time scales, including the adequacy of the model assumption of initial steady-state. Therefore, such experiments have a great potential in revealing possible inaccuracies, misrepresentations or missing processes in the model.

As an example, we ran a set of simulations of one such 2-hour isothermal evaporation followed by a standard heated desorption (Fig. 14). For the three pairs of panels, the same parameters were used as for Figs. 9-11, respectively, i.e. such that they would yield good fits of the $C_8H_{12}O_5$ thermogram in a standard run (thin green lines in Fig. 14; same as in Figs. 9-

11). The respective simulation results differ in terms of how much material has reached the CIMS after 2 hours of unheated evaporation (decrease in magenta lines), and in the effect of this period on the shape of the eventually obtained thermogram (thick vs. thin green lines after 120 min). The orange lines reveal that in all these cases an appreciable fraction of $C_8H_{12}O_5$ is simulated to be adsorbed to (filter) surfaces at the end of 120 min of unheated evaporation. That is because even though the (effectively) high-volatility components evaporate fairly rapidly from the aerosol particles, their assigned $C^*_0$ (0.3-2 μg m$^{-3}$) still lets them "stick" to the large filter surface area. Hence it passes on to the CIMS much more slowly, with corresponding consequences for the thermogram shape obtained upon eventual heating. This effect is smaller for a higher $C^*$, hence Approach 2 (Fig. 14B) is least affected. The biggest effect to the thermogram, when followed by a 2-hour isothermal evaporation, is seen for the reversible oligomer case (Fig. 14B), as most oligomers are dissociating to monomers within that 2-hour period at the dissociation rate obtained in section 5.1.3, $k_{d,0} = 5 \times 10^{-4}$ s$^{-1}$. Therefore, we would expect most of the thermogram shoulder to disappear. If experimental data shows that it does not, we will have to involve also more stable parent compounds to explain the full observations, or revise our assumption of initial steady-state (e.g., oligomers may still be forming after filter- sampling).

For actual results of experiments of the described type, plus application of our model, we refer to the forthcoming manuscript by D'Ambro et al. (2018).

**6 Conclusions**

We have developed and presented here a model that can be used, in general successfully, to reproduce thermograms obtained by FIGAERO-CIMS. The model is based on physical and chemical processes that are known or have been suspected to control thermal desorption of OA specifically, as well as OA properties more generally. Thus, by using the model to reproduce the thermograms of specific compositions quantitative information on the volatility and insights into the chemical environment can be derived. The model offers a high degree of flexibility. Several compounds can be readily implemented into single simulations, monomers can be formed by decomposition of parent compounds (treated as non-volatile) or reversibly enter and leave non-volatile states (such as oligomers).

The model reproduces very well the thermograms obtained from FIGAERO measurements for the test case used in this study: SOA derived from α-pinene dark ozonolysis. As observed previously, most compositions' thermograms feature a peak, but a large fraction of the respective material desorbs at higher than expected temperatures. The peak can typically be explained by a single desorbing compound, with the temperature of peak signal ($T_{max}$) largely controlled by the assigned room-temperature saturation concentration $C^*_0$, in agreement with previous thermogram calibrations and interpretations. The remaining high-temperature fraction of the thermograms, in our test case, typically constitutes >70% of the respective signal (Tables 2 and 3).

The majority of individual compositions' thermograms could be reproduced in the model by using only a single compound, defined by molecular weight, $C^*_0$ and $\varDelta H$, but assuming that at the beginning of desorption, it is in steady-state between a

free (to evaporate) state and a nonvolatile state, such as forming part of an oligomer. The model suggests that formation and dissociation of the oligomer state are typically governed by unexpectedly low activation energies, assumed negligible for the formation reaction and mostly < 25 kJ mol$^{-1}$ for dissociation. Room-temperature dissociation rates are on the order of ~ 1 hr$^{-1}$, in agreement with overall SOA evaporation time-scales reported in previous studies (e.g., Vaden et al., 2011; Trump and

Donahue, 2014; Roldin et al., 2014; Kolesar et al., 2015a). The low activation energies in our simulations suggest that the bulk of large monomers that desorbs from α-pinene SOA is initially bound in matrices joined by a network of non-covalent H bonds. Our assumption that this binding process is inherently reversible has actually little effect on the quality of our fits, but implies formation time scales < 15 min.

On the other hand, the thermograms of some smaller observed oxygenated organic compositions (e.g. $C_2H_2O_3$) are explained

best by modeling them as *only* being the product of thermal decomposition, consistent with the expectation that such small compounds would not actually be present in the particle phase due to their high vapor pressure. These simulations use plausible activation energy distributions around $E_d \sim 100$ kJ mol$^{-1}$, values suggestive of peroxide (O–O) bond cleavage. Cleavage of such stronger bonds may also be involved in the thermograms of the larger compositions, and a clear contribution indeed appears likely for other FIGAERO data (Lopez-Hilfiker et al., 2015), although a rather slow isothermal

evaporation at maximum temperature (200 °C) for those cases appears most consistent with lower bond energies.

Overall, considerably more than 50% of the oxygenated organic material that is observed to desorb has been produced by decomposition processes. This conclusion conforms with conclusions reached also through less involved analysis of FIGAERO data (e.g., Lopez-Hilfiker et al., 2015). Although based on a small selection of data, these numbers also broadly agree with earlier studies that have used other methods to try to quantify how large of a fraction of monoterpene-derived

SOA consists of oligomeric compounds. It has been concluded that these compounds constitute around 50% of SOA mass in laboratory setups (Baltensperger et al., 2005; Hall and Johnston, 2011; Putman et al., 2012), or even well above 50% (Gao et al., 2004; Kolesar et al., 2015a), in general agreement with observations made on monoterpene-dominated ambient SOA (Kourtchev et al., 2016).

Still, many individual thermograms feature a prominent main peak, typically at relatively low temperature. The value of this

temperature ($T_{max}$) is primarily controlled by $C^*_0$ and indirectly by $\Delta H$, in agreement with the $T_{max}$-$C^*_0$ relationships commonly used in existing literature to interpret FIGAERO thermograms (e.g., Mohr et al., 2017; Stark et al., 2017; Huang et al., 2018). Yet, this interpretation alone will result in a misleading description of the investigated SOA, as the properties of more than half of the observed organic material are implicitly misinterpreted. Thermograms entirely shaped by decomposition processes, such as that of $C_2H_2O_3$, would be misinterpreted altogether. This study is not the first to point that

out but it illustrates the issue in detail, and our model now offers a tool that can be used to analyze the majority component of organic mass that is not described by $T_{max}$ alone. Our model treats this material as non-volatile, but it is sufficient for its constituents to have only such low volatility that their decomposition occurs at lower temperatures than their evaporation. So in addition to fractions of that low-volatility material in SOA, we are now able to provide well-defined upper limits to their volatility. A more complete analysis of the FIGAERO datasets, aided by our model, is clearly warranted. The model can also

serve as a handle on predicting the effects of various potential experimental variables on FIGAERO observations. E.g., effects of varying particle viscosity, filter loading, ramp rates, etc., can be examined.

The flexibility of the current model may also be considered a weakness, as there can be several solutions to reproducing a certain thermogram. Ultimately, such ambiguities are resulting from the lack of detailed understanding of the processes underlying SOA properties, plus instrumental uncertainties, the very issues the model is meant to investigate. It also shows that the standard FIGAERO desorption experiments alone are insufficient for providing the necessary information for constraining all free parameters simultaneously. Consequently, we have discussed possible future experiments, such as composition-resolved isothermal evaporations, that could be designed to address these issues. It may also prove crucial to obtain a better understanding of blank experiments, and consequently on the possible errors when subtracting background based on those experiments. Comparison with results from detailed aerosol chemistry and mass transport models (e.g., Shiraiwa et al., 2012; Roldin et al., 2014) may further prove useful in improving our overall descriptive and predictive accuracies.

**Code Availability**

A documented version of the model's MATLAB code is hosted on GitHub and publicly available at github.com/sschobes/FIGAERO_model. Links to that repository, as well as to possible future locations of the code, will be provided on the University of Washington (Thornton Lab) and University of Eastern Finland (Aerosol Physics Group) websites.

**Acknowledgments**

We thank Olli-Pekka Tikkanen (University of Eastern Finland) for useful and on-going discussions on model optimization, and support in publishing the model code. Dale R. Durran (University of Washington), Kari E. J. Lehtinen (University of Eastern Finland), Neil M. Donahue (Carnegie Mellon University) and Paul J. Ziemann (University of Colorado) are acknowledged for useful comments regarding model development and interpretation of results, and Daniel Schlesinger (Stockholm University) for comments on the manuscript. Support for S.S. by the Academy of Finland (grants no. 272041 and no. 310682) and for E.L.D. by the National Science Foundation Graduate Research Fellowship (grant no. DGE-1256082) is gratefully acknowledged.

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

**Table 1:** Model-derived position of signal peak in temperature space, ($T_{max}$) for pure-compound particles and ranges of volatilities $C^*_0$ and particle sizes $D_{P,0}$ (cf. Fig. 5), and for vapor-surface interactions included vs. excluded. Other parameters are as for Figs. 5 and 6.

| $C^*_0$ ($\mu$g m$^{-3}$) | $D_{P,0}$ (nm) | $T_{max}$ (°C) | |
| --- | --- | --- | --- |
| | | $\tau C_W = 8.77$ mg m$^{-3}$ (default) | No vapor-surface interactions |
| 1 | 5 | 56 | < 25 |
| | 50 | 57 | 36 |
| | 150 | 57 | 43 |
| | 500 | 58 | 52 |
| $10^{-1}$ | 5 | 71 | 37 |
| | 50 | 71 | 49 |
| | 150 | 71 | 56 |
| | 500 | 73 | 66 |
| $10^{-2}$ | 5 | 87 | 51 |
| | 50 | 87 | 63 |
| | 150 | 87 | 70 |
| | 500 | 88 | 81 |
| $10^{-3}$ | 5 | 104 | 65 |
| | 50 | 104 | 78 |
| | 150 | 104 | 86 |
| | 500 | 106 | 97 |
| $10^{-4}$ | 5 | 123 | 81 |
| | 50 | 123 | 95 |
| | 150 | 123 | 102 |
| | 500 | 124 | 114 |
| $10^{-5}$ | 5 | 143 | 97 |
| | 50 | 143 | 111 |
| | 150 | 143 | 121 |
| | 500 | 144 | 134 |
| $10^{-6}$ | 5 | 165 | 115 |
| | 50 | 165 | 130 |
| | 150 | 165 | 141 |
| | 500 | 166 | 155 |

**Table 2:** Free parameters used for model runs in Figs. 9-11, for reproducing the $C_8H_{12}O_5$ thermogram obtained from an α-pinene ozonolysis (dark) SOA experiment. Particle size of 197 nm was used, the evaporation coefficient was always $\alpha = 1$. The Raoult term at start of desorption ($\chi_0$) was based on observed signals (see text). Oligomerization was modeled reversibly in case of Fig. 11 (last line), indicated by a value provided for $k_{g,0}$ and an initial oligomer fraction determined by assumed steady state initially. The activation energy for oligomerization was $E_{g,0} = 0$ kJ mol$^{-1}$.

| Measured: | | Ref: | Model runs: | | | | | | |
| Composition | $\chi_0$ | Fig. | Fraction | Evaporation | | Oligomer dissociation (and formation) | | | |
| | | | | $C^*_0$ (µg m$^{-3}$) | $\Delta H$ (kJ mol$^{-1}$) | $k_{g,0}$ (s$^{-1}$) | $k_{d,0}$ (s$^{-1}$) | Initially oligomer | $E_d$ (kJ mol$^{-1}$) |
|---|---|---|---|---|---|---|---|---|---|
| $C_8H_{12}O_5$ | $3.7 \times 10^{-2}$ | 9 | 0.39 | $3 \times 10^{-1}$ | | - | - | 0 | - |
| | | | 0.2 | $3 \times 10^{-2}$ | | - | - | 0 | - |
| | | | 0.14 | $3 \times 10^{-3}$ | 105 | - | - | 0 | - |
| | | | 0.14 | $3 \times 10^{-4}$ | | - | - | 0 | - |
| | | | 0.14 | $3 \times 10^{-5}$ | | - | - | 0 | - |
| | | 10 | 0.27 | | | - | $3 \times 10^{-4}$ | 1 | 80 |
| | | | 0.16 | | | - | $4 \times 10^{-5}$ | 1 | 85 |
| | | | 0.14 | | | - | $5 \times 10^{-6}$ | 1 | 90 |
| | | | 0.1 | 2 | 80 | - | $7 \times 10^{-7}$ | 1 | 95 |
| | | | 0.08 | | | - | $9 \times 10^{-8}$ | 1 | 100 |
| | | | 0.08 | | | - | $1 \times 10^{-8}$ | 1 | 105 |
| | | | 0.08 | | | - | $2 \times 10^{-9}$ | 1 | 110 |
| | | | 0.08 | | | - | $2 \times 10^{-10}$ | 1 | 115 |
| | | 11 | 1 | 0.8 | 88 | $2 \times 10^{-3}$ | $5 \times 10^{-4}$ | 0.77 | 6 |

**Table 3:** Free parameters used for the model runs in Figs. 12 and 13, analogous to Table 2.

| Measured: | | Ref: | Model runs: | | | | | | |
| --- | --- | --- | --- | --- | --- | --- | --- | --- | --- |
| | | | | Evaporation | | Oligomer dissociation (and formation) | | | |
| Composition | $\chi_0$ | Fig. | Fraction | $C^*_0$ ($\mu$g m$^{-3}$) | $\Delta H$ (kJ mol$^{-1}$) | $k_{g,0}$ (s$^{-1}$) | $k_{d,0}$ (s$^{-1}$) | Initially oligomer | $E_d$ (kJ mol$^{-1}$) |
| **C$_8$H$_{10}$O$_5$** | $2 \times 10^{-2}$ | 12A | 1 | 4 | 60 | $2 \times 10^{-3}$ | $2 \times 10^{-4}$ | 0.89 | 17 |
| **C$_{10}$H$_{14}$O$_5$** | $1.3 \times 10^{-2}$ | 12B | 0.18 | 0.3 | 115 | - | - | 0 | - |
| | | | 0.82 | 0.08 | 105 | $1 \times 10^{-2}$ | $7 \times 10^{-4}$ | 0.93 | 6 |
| **C$_2$H$_2$O$_3$** | $6 \times 10^{-3}$ | 12C | 0.02 | Not limiting | | - | $1 \times 10^{-3}$ | 1 | 76 |
| | | | 0.06 | | | - | $2 \times 10^{-4}$ | 1 | 81 |
| | | | 0.18 | | | - | $3 \times 10^{-5}$ | 1 | 86 |
| | | | 0.18 | | | - | $3 \times 10^{-6}$ | 1 | 91 |
| | | | 0.17 | | | - | $5 \times 10^{-7}$ | 1 | 96 |
| | | | 0.11 | | | - | $6 \times 10^{-8}$ | 1 | 101 |
| | | | 0.08 | | | - | $8 \times 10^{-9}$ | 1 | 106 |
| | | | 0.07 | | | - | $1 \times 10^{-9}$ | 1 | 111 |
| | | | 0.06 | | | - | $1 \times 10^{-10}$ | 1 | 116 |
| | | | 0.06 | | | - | $2 \times 10^{-11}$ | 1 | 121 |
| **C$_{18}$H$_{28}$O$_6$** | $6 \times 10^{-4}$ | 13A | 1 | 0.25 | 60 | - | - | 0 | - |
| | | 13B | 1 | Not limiting | | - | $2 \times 10^{-6}$ | 1 | 92 |
| | | 13C | 0.23 | 0.02 | 120 | - | - | 0 | - |
| | | | 0.45 | $1 \times 10^{-3}$ | 133 | - | - | 0 | - |
| | | | 0.32 | $5 \times 10^{-5}$ | 146 | - | - | 0 | - |

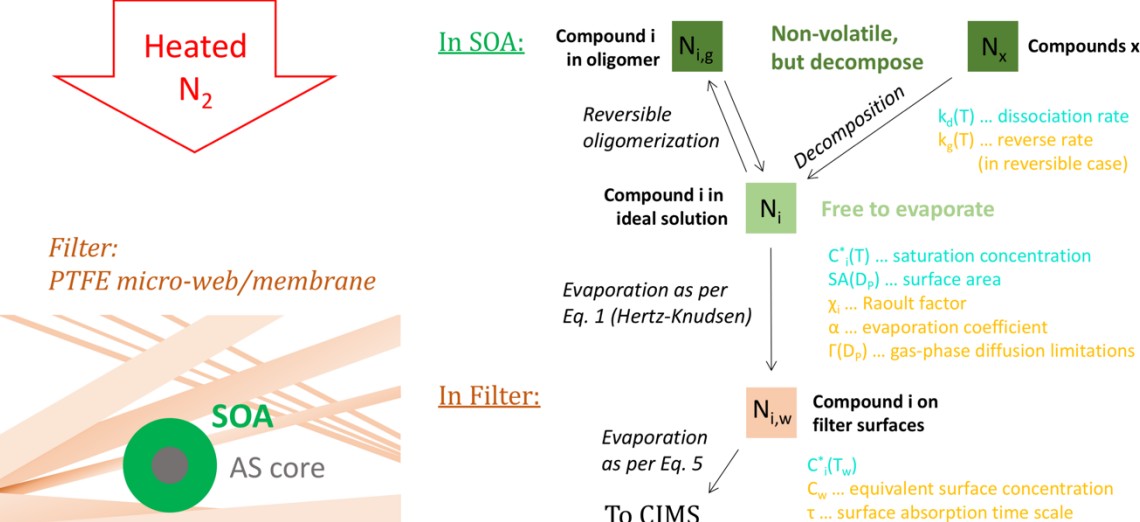

**Figure 1: Schematic of the processes implemented in our model. The left-hand side drawing gives an impression of the overall situation: An SOA particle (green), in this case with a core from an ammonium sulfate (AS) seed particle (gray), is deposited on the FIGAERO collection filter and exposed to a heated flow of $N_2$. The core of the filter is a microporous membrane composed of a network of PTFE fine fibers (a.k.a. fibrils; beige). These fibrils are not accurately depicted here, the drawing is rather supposed to convey that the deposited particles are likely nested inside a complex network of fibrils that provide a large total surface area. The right-hand side summarizes the processes that are simulated for molecules of a certain compound $i$ ($N_i$). Included is a list of factors that chiefly control these processes: Factors contributing to evaporation are colored cyan, factors inhibiting evaporation are colored orange.**

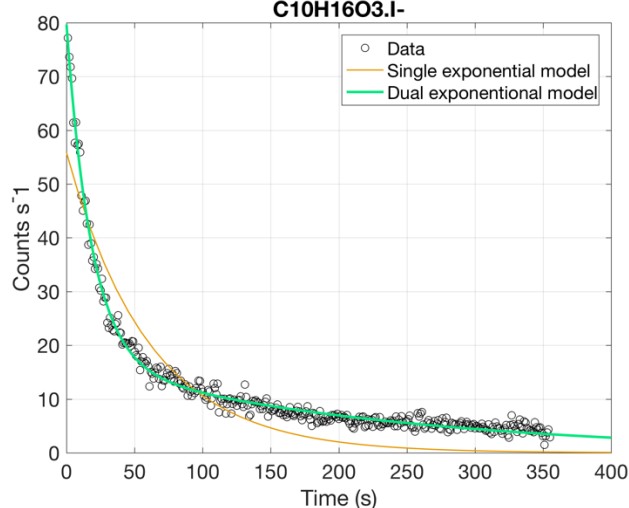

**Figure 2: Decay of the signal for pinonic acid (detected as $C_{10}H_{16}O_3.I^-$) during one of the experiments comprising a blank aerosol collection period, i.e. with a particle filter in the sampling line, followed by desorption in room temperature. The dotted line is the fit obtained using Eq. (8); the solid line fit was obtained using Eq. (9).**

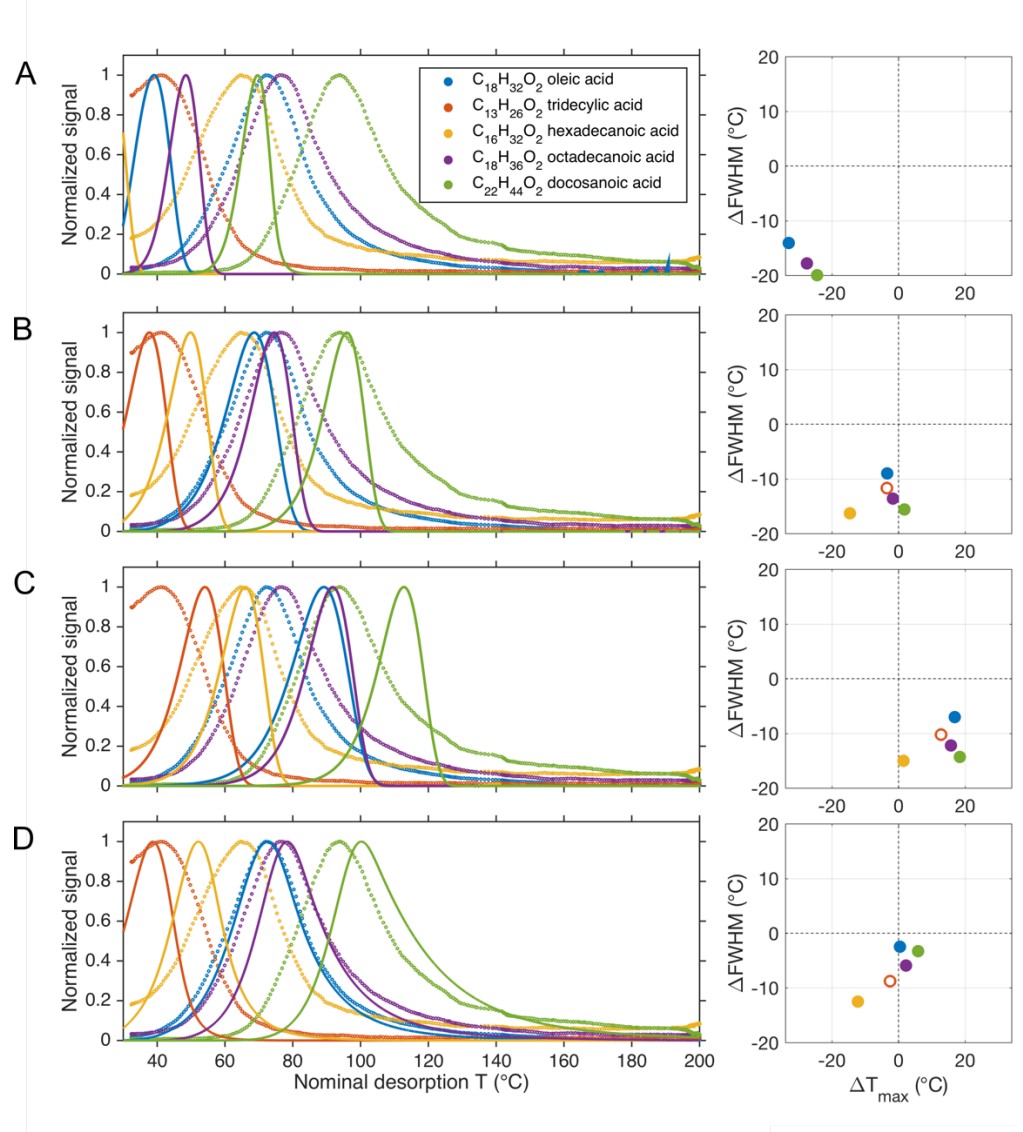

**Figure 3: Comparison of experimental results from depositing a solution containing monocarboxylic acids (Lopez-Hilfiker et al., 2014) with four different model results. The left-hand panels show the measured (circles) and modeled (lines) thermograms; the right-hand panels summarize differences between model and experiment regarding peak position ($\Delta T_{max} = T_{max, mod} - T_{max, exp}$) and full width at half maximum ($\Delta FWHM = FWHM_{mod} - FWHM_{exp}$). For panels (A), vapor-surface interactions after initial desorption were excluded in the model. For subsequent panels, these interactions were included as per Eqs. (5) and (6), using a wall parameter $\tau C_w$ of 8.77 mg m$^{-3}$ s (B) or 149 mg m$^{-3}$ s (C). In panels (D), the wall parameter was the same as in (B), but assuming uneven desorption temperatures across the deposit, as described in the text and in Fig. 4.**

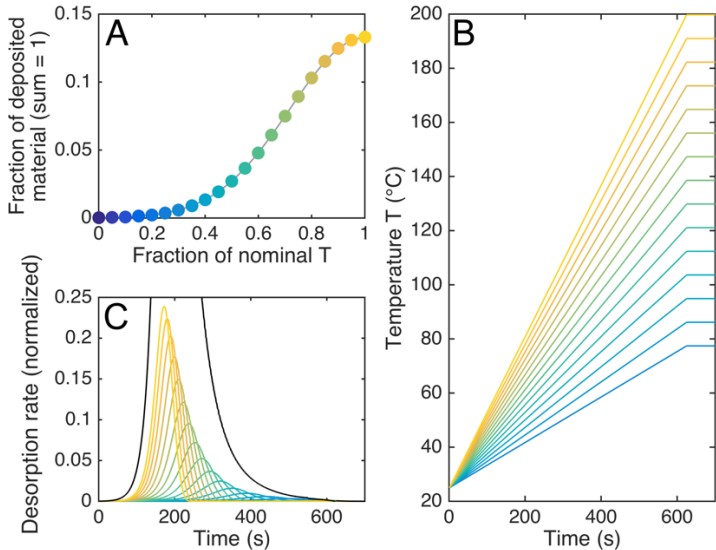

**Figure 4: Illustration of the assumptions behind the model results in Fig. 3C. Panel (A) shows the fractions of deposited material (on the ordinate) that were each assumed to be exposed to a fraction of the nominal desorption temperature (on the abscissa). This function is Gaussian with a standard deviation of 0.28. Panel (B) shows the respectively assumed temperature profiles, except for the lowest six that we neglected. Panel (C) shows the respective desorption rates as a function of time, and also the sum of all rates (black; peaking at 1), illustrating how the assumptions here lead to a tail in the sum thermogram (cf. Fig. 3D). In all panels, the color scheme reflects the maximum desorption temperature for each fraction or profile, from 200 °C (lightest yellow) to 79 °C (darkest blue in panels B-C) or 25 °C (darkest blue in panel A).**

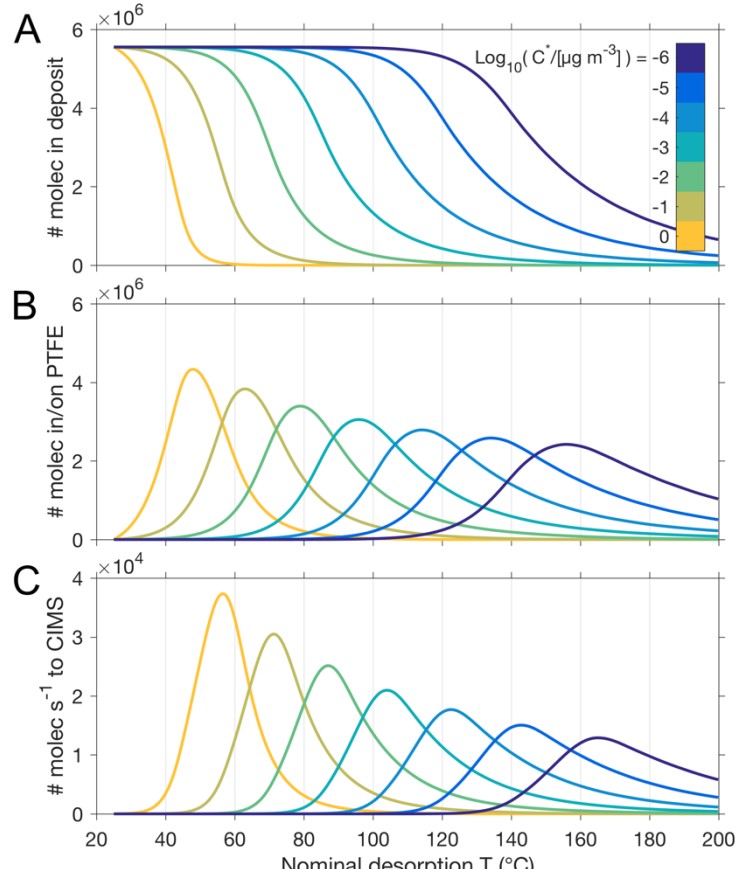

**Figure 5: Illustration of model outputs for the simple system of a 150-nm aerosol particle composed of only one compound, for seven different values of this compound's saturation vapor concentration at room temperature (25 °C) $C^*_0$. Panel (A) shows the number of molecules remaining in the particle as a function of desorption temperature ($T$), which is ramped at a constant rate from 25 to 200 °C and hence proportional to time. Panel (B) shows the number of molecules that have evaporated from the particle and are modeled to stick on surfaces prior to entering the CIMS at the rate shown in panel (C).**

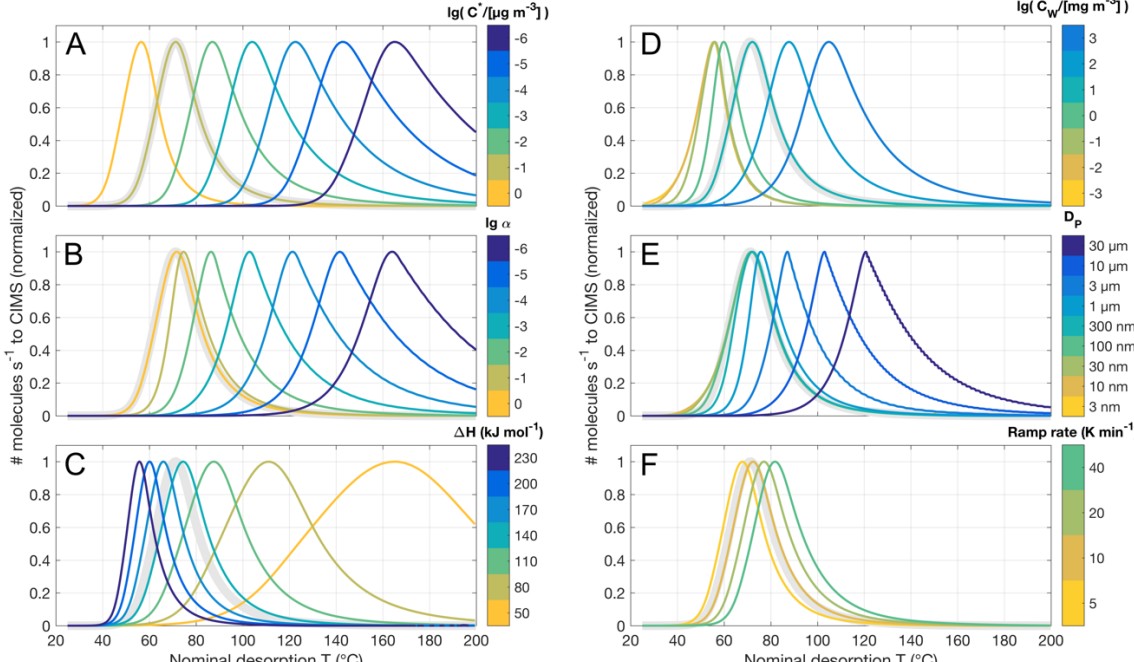

**Figure 6: Normalized model thermograms for the simple system of an aerosol particle composed of only one compound, varying a certain input parameter for each panel. The default parameters are: $C^*_0 = 0.1$ µg m$^{-3}$, $\alpha = 1$, $\Delta H = 150$ kJ mol$^{-1}$, $\tau C_W = 8.77$ mg m$^{-3}$, $D_{P,0} = 150$ nm, temperature ramp rate = 0.14 K s$^{-1}$; the corresponding default thermogram is shown in bold light gray in each panel. Panel (A) is the same as Fig. 5C, i.e. varying $C^*_0$, except that each thermogram is normalized to one. Panel (B) shows varying the evaporation coefficient from $\alpha = 1$ down to $10^{-6}$, panel (C) the vaporization enthalpy $\Delta H$ between 50 and 230 kJ mol$^{-1}$, panel (D) the wall "stickiness" $C_W$ between 0.1 and 1000 mg m$^{-3}$, and panel (E) the initial particle diameter $D_{P,0}$. Panel (F) shows the effect of adjusting the temperature ramp rate.**

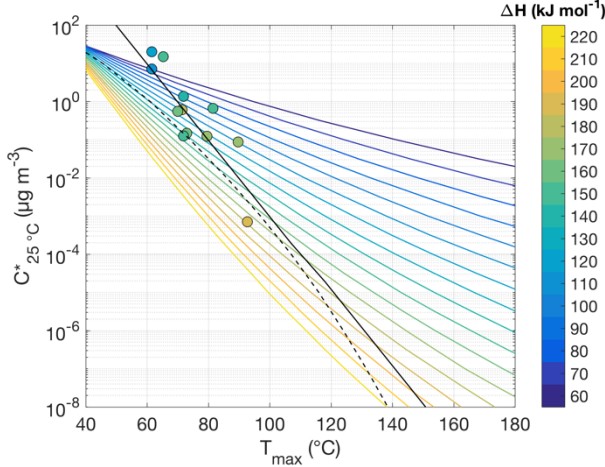

**Figure 7: The relationship between $T_{max}$ (abscissa) and both the saturation concentration $C^*$ at room temperature (ordinate) and the vaporization enthalpy $\Delta H$ (color scheme). Results from model simulations are summarized by the colored lines. Typical assumptions and parameters were used: $\alpha = 1$, $\tau C_W = 8.77$ mg m$^{-3}$, $D_{P,0} = 200$ nm, temperature ramp rate $= 0.14$ K s$^{-1}$. Experimental observations by Lopez-Hilfiker et al. (2014) are shown as colored circles, their fit by Mohr et al. (2017) is shown as black line. The dashed black line depicts the semi-empirical $C^*$-$\Delta H$ relation developed by Epstein et al. (2010): $\Delta H = 131 - 11 \log_{10}(C^*)$.**

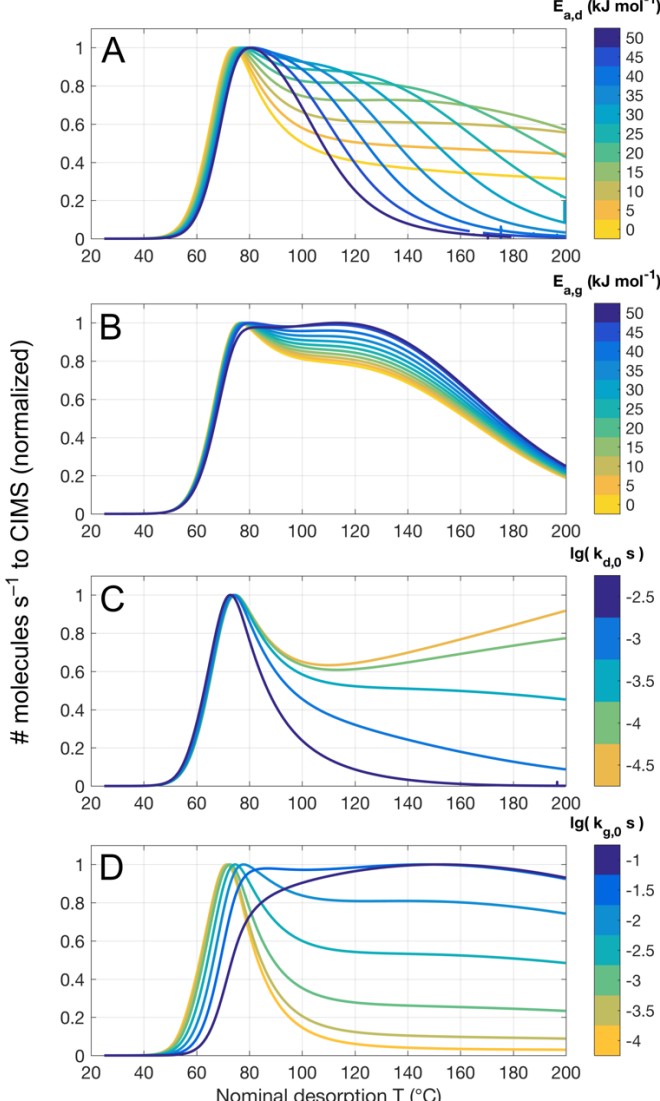

**Figure 8: Sample sensitivity tests for the four parameters controlling oligomer formation and dissociation. Same model parameters are used as for sensitivity tests shown in Fig. 6, but including oligomerization (in the forward direction, denoted by subscript $g$, and reverse direction, denoted by subscript $d$). Used default values, unless noted otherwise, are: $E_d = 10$ kJ mol$^{-1}$, $E_g = 0$ kJ mol$^{-1}$, $k_{g,0} = 3 \times 10^{-4}$ s$^{-1}$, $k_{d,0} = 3 \times 10^{-4}$ s$^{-1}$. For Panel A, $E_d$ varies between 0 and 50 kJ mol$^{-1}$, while $E_g$ is 25 kJ mol$^{-1}$, and vice versa for Panel B ($E_d = 25$ kJ mol$^{-1}$, $E_g = 0$ to 50 kJ mol$^{-1}$). Panels C and D show variations in $k_{d,0}$ from $3 \times 10^{-5}$ to $3 \times 10^{-3}$ s$^{-1}$, and $k_{g,0}$ from $10^{-4}$ to $10^{-1}$ s$^{-1}$, respectively.**

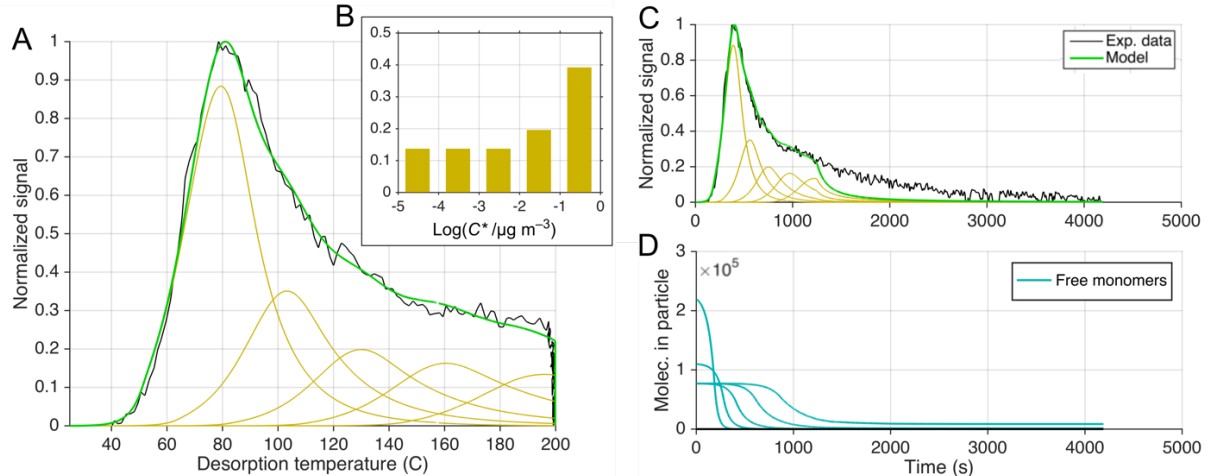

**Figure 9: Illustration of Approach 1 (out of three) for reproducing the thermogram observed for composition $C_8H_{12}O_5$ during desorption of SOA generated from dark α-pinene ozonolysis. This approach is based on excluding oligomerization but using five logarithmically spaced VBS bins at $C^*_0 = 3 \times 10^{-1}$ to $3 \times 10^{-5}$ µg m$^{-3}$, $\Delta H = 105$ kJ mol$^{-1}$ and $\alpha = 1$. Assumed relative abundances for each bin are shown in panel (B). Panel (A) is the classical normalized thermogram, i.e. in temperature space. Experimental data is shown in black, model data in beige for the individual bins and in green for their sum. Panel (C) presents the thermogram in time ($t$) space, which is identical to panel (A) up to $t = 1250$ s, because temperature was ramped linearly with time to 200 °C, but revealing a failure in reproducing the soak period (constant 200 °C for $t > 1250$ s). In panel (D), the number of molecules still in the particle phase is plotted vs. time, one line for each VBS bin.**

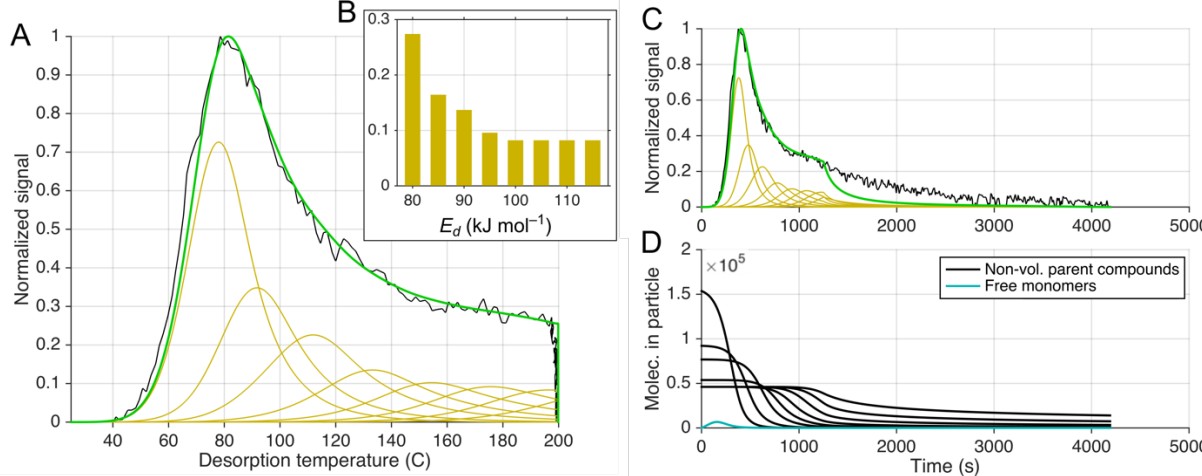

Figure 10: Equivalent to Fig. 9, but presenting the results of using Approach 2 (out of three) for reproducing the observed thermogram. All $C_8H_{12}O_5$ molecules are formed from irreversible thermal decomposition of non-volatile material, following the Arrhenius relation (see text) using a set of eight activation energies ($E_d$), ranging from $E_d = 80$ to 115 kJ mol$^{-1}$. Their subsequent evaporation is modeled using $C_0^* = 2$ µg m$^{-3}$, $\Delta H = 80$ kJ mol$^{-1}$ and $\alpha = 1$, i.e., practically not limiting. Assumed relative contributions of each $E_d$ are shown in panel (B). (Note the consequently different units for the abscissa in panel (B) compared to Figs. 9 and 11.) Panel (D) shows the abundances of the corresponding non-volatile parent compounds still in the particle vs. time (black).

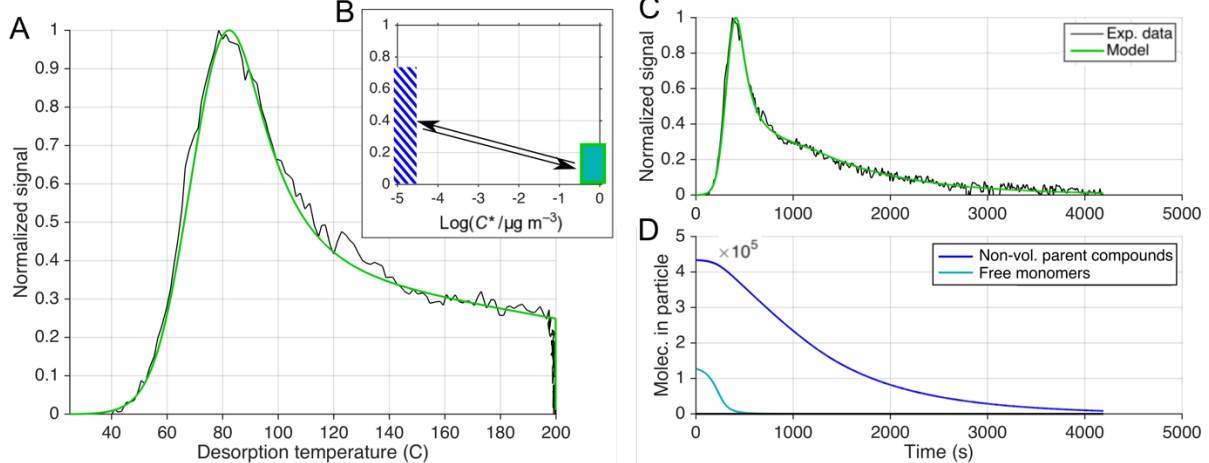

**Figure 11:** Equivalent to Fig. 9, but presenting the results of using Approach 3 (out of three) for reproducing an observed thermogram. Here, reversible oligomerization is included as in Eq. (11), and $E_d$ is independent from $k_{d,0}$ (Eq. (12)). Only one compound is used, with $C^*_0 = 0.8$ $\mu$g m$^{-3}$, $\Delta H = 88$ kJ mol$^{-1}$ and $\alpha = 1$. Oligomerization parameters are $k_{g,0} = 1.7 \times 10^{-3}$ s$^{-1}$, $k_{d,0} = 5 \times 10^{-4}$ s$^{-1}$, $E_g = 0$ kJ mol$^{-1}$, $E_d = 6$ kJ mol$^{-1}$. The initial fraction of $C_8H_{12}O_5$ in the oligomer state is $k_{g,0}/(k_{g,0}+k_{d,0}) = 77\%$. Panel (D) illustrates how $C_8H_{12}O_5$ is released from decomposing oligomers during particle desorption (dark blue). In panel (B), the low-volatility (presumably oligomer) state is represented by a blue-striped bar at close to $10^{-5}$ $\mu$g m$^{-3}$. However, that is only a rough upper limit, whereas the actual volatility is not known. The arrows illustrate the modeled on-going reversible oligomerization reactions between monomers (modeled with a volatility of 0.8 $\mu$g m$^{-3}$) and practically nonvolatile oligomers.

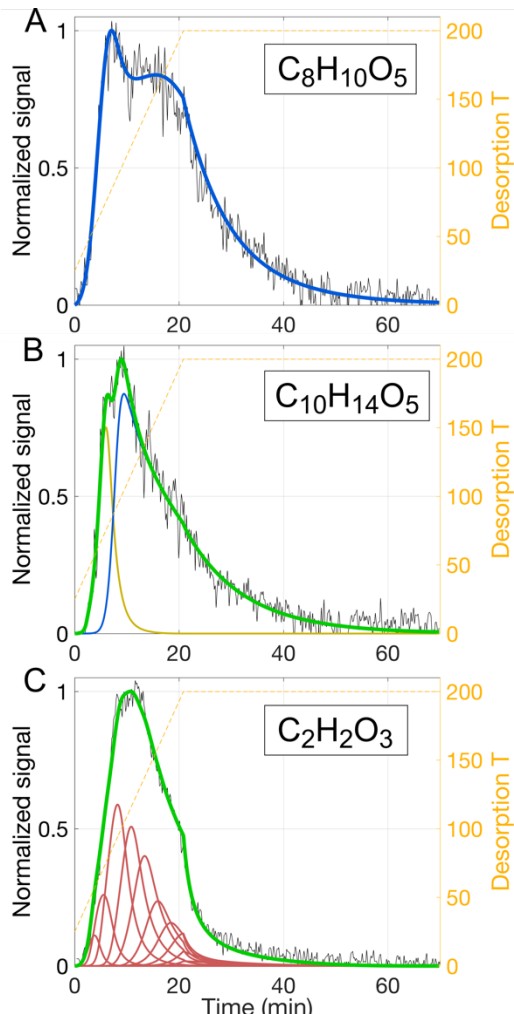

**Figure 12: Examples of thermogram reproductions for three different compositions: $C_8H_{10}O_5$ (panel A), $C_{10}H_{14}O_5$ (panel B), $C_2H_2O_3$ (panel C). All data are from the same aerosol desorption as data used for Figs. 9-11 ($C_8H_{12}O_5$). Experimental data is normalized and plotted in black; desorption temperature is shown as orange dashed lines (right-hand ordinates). Model results are presented as colored solid lines, the color showing the source of the modeled signal: Beige for simple single-compound evaporation, blue when reversible oligomerization is included and $E_d$ is independent from $k_{d,0}$, and red for unidirectional thermal decomposition and fixed relation between $E_d$ and $k_{d,0}$ (Eq. (18)). Sums are drawn in green. The used model parameters are given in Table 3.**

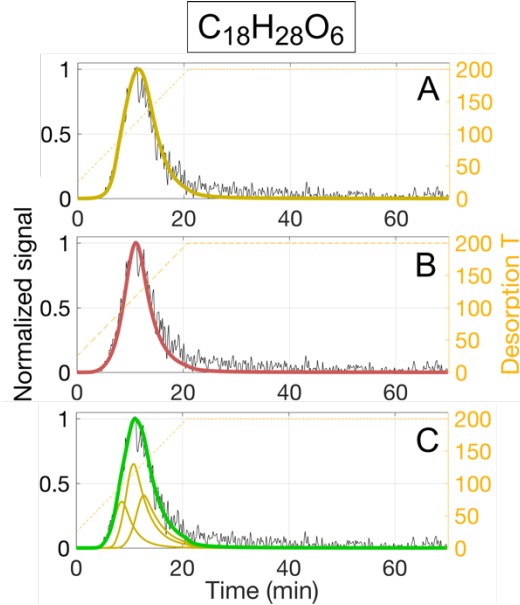

**Figure 13: Three different ways of modeling the thermogram from $C_{18}H_{28}O_6$, assuming either direct evaporation of a single compound (panel A), or evaporation limited by thermal decomposition into $C_{18}H_{28}O_6$ (panel B), or the direct evaporation of three isomers (panel C). Data source, normalization, as well as the color coding are the same as in Fig. 12, and the used model parameters are included in Table 3.**

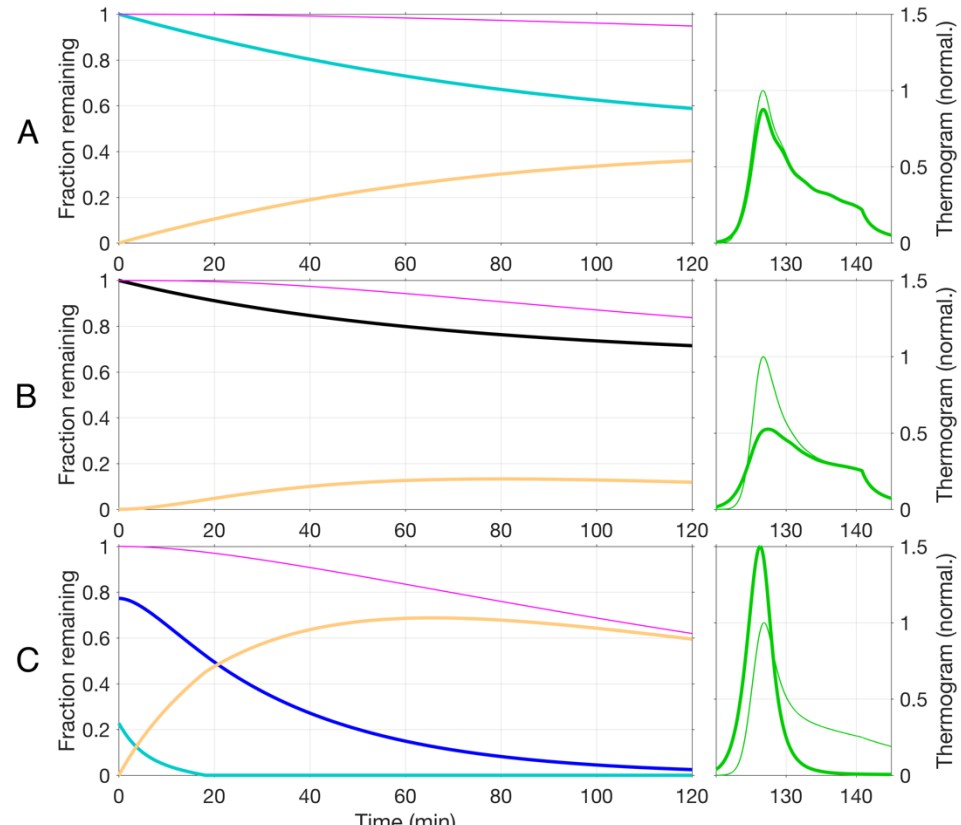

**Figure 14: Simulations of a 2-hour period of isothermal evaporation of $C_8H_{12}O_5$ at room temperature (left-hand panels), and the subsequent thermograms (right-hand panels). Input parameters are identical to those for Figs. 9 (panels A), 10 (panels B) and 11 (panels C). The thick lines in the left-hand panels show the fractions of monomers remaining in various states as indicated by the**

5 **color scheme. Light blue fractions (panels A and C) are monomers in the particle and free to evaporate, the dark blue fraction (panel C) is monomers bound in a non-volatile (oligomer) state initially in steady-state with the free monomers, the black fraction (panel B) are monomers in a pre-defined distribution of thermally decomposing non-volatile compounds (black), and the orange fractions are monomers already desorbed from the particle but adsorbed to surfaces. The thin magenta lines are the respective sums, equal to the fraction of monomers not yet sampled by the CIMS. The right-hand panels show the simulated thermograms**

10 **obtained following the 2-hour isothermal evaporation period (thick green lines), normalized to the maximum of the thermogram obtained without the 2-hour period (shown by the thin green line).**