# Peer review of "A model framework to retrieve thermodynamic and kinetic properties of organic aerosol from composition-resolved thermal desorption measurements"

_Atmospheric Chemistry and Physics, 2018_

## Referee Comment (RC1) · Anonymous Referee #1 · 12 Jun 2018

The manuscript by Schobesberger et al. entitled, "A model framework to retrieve thermodynamic and kinetic properties of organic aerosol from composition-resolved thermal desorption measurements" is a wonderful addition to the discussion on the thermodynamic and kinetic properties of secondary organic aerosol. The model framework is described in detail and the analysis of how each fit parameter affects the shape of the desorption curve is useful. Since the best model fit was achieved by characterizing the SOA as being mostly composed of oligomeric-like molecules with parameters that agree well with previous studies, the current manuscript is moving the commu-

nity closer to a robust set of parameters for describing the thermodynamic and kinetic properties of SOA.

Minor revisions suggested: Page Line 6 – "related measures of volatility" are mentioned. Could the authors expand on this? In Section 3.7 running a distribution of particle sizes is said to be too computationally expensive. Since this is the case, I would suggest size selecting particles during the experiment so that this source of uncertainty is minimized. Page 16 line 23 – what was the geometric standard deviation of the particle distribution? Page 18 line 16-17: The sentence would read better if "of which" was after the "," and before "the relatively slow decomposition" Page 18 line 19: needs to be "(see also section 4.5)" Page 20 line 3: should be "We used the model" Page 21 line 5 (Section 5.3): Could you please restate the initial values obtained with alpha = 1? Page 23 line 1: should be "convincing and suggests that we are able to attribute" Figure 4: The plots are mislabeled: C should be B and B should be C. The summed trace in what is currently plot 4b is truncated, please give the maximum value of the curve in the caption. Figure 8: should be "A particle size of 197 nm corresponds to the volume median diameter" Figure 9: It would be good to keep the colors consistent between Figures 8 and 9. There are numerous typos, please proofread. What does "N.B." stand for? Add that the black traces are non-volatile to the legend for panel D.

1. Does the paper address relevant scientific questions within the scope of ACP? Yes, the paper discusses experiments and a model framework that aim to further our understanding of the mechanisms of particle evaporation. 2. Does the paper present novel concepts, ideas, tools, or data? Yes, the model framework, supporting experiments, and uncertainty analysis are all novel contributions. The main experiment, the analysis of SOA formed from the dark ozonolysis of alpha-pinene with FIGAERO CIMS, is an important expansion on previous experiments. 3. Are substantial conclusions reached? The quantification of volatility parameters using the model fits to experimental data are substantial conclusions of this manuscript. 4. Are the scientific

methods and assumptions valid and clearly outlined? The model framework and all assumptions and sources of uncertainty are described in great detail. 5. Are the results sufficient to support the interpretations and conclusions? The model runs clearly show good agreement with the experimental results and thus support the conclusion that the model parameters describe the thermodynamic and kinetic properties of the system. 6. Is the description of experiments and calculations sufficiently complete and precise to allow their reproduction by fellow scientists (traceability of results)? Yes, the descriptions are all very thorough. 7. Do the authors give proper credit to related work and clearly indicate their own new/original contribution? Yes, previous work is cited where needed and compared to current results. 8. Does the title clearly reflect the contents of the paper? Yes 9. Does the abstract provide a concise and complete summary? The abstract is a little heavy on describing the methods. More of the abstract could be devoted to the conclusions and best-fit model parameters. 10. Is the overall presentation well structured and clear? Yes 11. Is the language fluent and precise? Yes 12. Are mathematical formulae, symbols, abbreviations, and units correctly defined and used? Yes 13. Should any parts of the paper (text, formulae, figures, tables) be clarified, reduced, combined, or eliminated? Yes, there is no explanation for the color scheme in Figure 4. Additionally, the panels are mislabeled in Figure 4. The traces in Figure 8D all appear to be the same color while a legend indicates there should be different colors based on volatility. Figure 9D should have the same color scheme as Figure 8D. The caption for Figure 9 does not read well; it seems as though text became jumbled. Figure 13 is difficult to interpret and needs some more explanation. In general, the axes need to be adjust to show the top portion of the graph, or information given in the caption as to the maximum value when it cannot be seen. 14. Are the number and quality of references appropriate? Yes 15. Is the amount and quality of supplementary material appropriate? Yes

---

## Referee Comment (RC2) · Anonymous Referee #2 · 13 Jun 2018

The authors present a detailed approach to modeling thermal desorption from a PTFE filter as is found in the FIGAERO CIMS inlet. This model is itself useful in the field, as that instrument is seeing wide use, and the authors further go on to explore properties of SOA, particularly with respect to volatility, kinetics, and potential thermal decomposition. The manuscript is well written and thorough, and represents a clear advance of knowledge. I recommend publication after addressing the comments below, which are largely minor and technical in nature.

Broader comments:

[Figure]

I appreciate the explicit discussion of what is (and is not) meant by "oligomerization" throughout this manuscript. Quite often this term is thought of (perhaps incorrectly, but nevertheless) as implying covalent homo-oligomers, such as dimers and trimers, while there is some evidence and reason to believe that SOA is substantially more complicated than that (e.g. methods able to measure dimers often do not see enough to explain partitioning). While the term is used throughout, this clarity is not brought forward until page 10. I recomment that to address the above conceptions the discussion of "oligomer" be brought forward to the introduction. It is also the reason I tend to prefer "accretion products" as the more universal term, but understand if the authors prefer to stick with the more common term "oligomers".

Throughout the manuscript, the authors sometimes mention the possibility that an ion may represent multiple isomers, but its not always clear to me to what extent this is being considered. Figure 11 demonstrates that there are many possible ways to fit each thermogram, and one could imagine for instance an ion consisting of monomers with a range of volatilies, and each also formed from one or two different oligomers. Panel B shows such an example a case in which two isomers are invoked to describe the thermogram, one which is pure and another comprised of low- and high-volatility components. While captured here, it may be a case existing in many of the observed ions, but to which the model is blind given its large number of free parameters. It seems here that in most cases there is a simplifying assumption that each ion can be treated as one compound (except in the case of 11B), which may or may not be a robust assumption. Trying to tackle this question may go beyond the scope of this manuscript, but it should probably be discussed more explicitly and added to the list of "challenges" in Section 5.

Technical comments:

Page 3 line 10: A word seems to be missing in "Other methods by"

Page 3 line 17: What do the author's mean by the "compositions of these molecules"?

I presume they mean molecular formulas?

Page 10 lines 18-22:

Page 10 line 27: The use of "compound" here and throughough is 'iffy', as for most practical FIGAERO applications a given ion may represent a mixture of multiple compounds. See comment above.

Page 11 line 18: It's not clear to me how k_g,0/(k_g,0+k_d,0) is calculated before each model run

Page 12 line 7-9: My understanding of both cited references is that FIGAERO CIMS saw half the mass, but also assumed equal sensitivity. However, when Isaacman-VanWertz et al. (Nature Chemistry, 2018, doi:10.1038/s41557-018-0002-2) applied the calibration approaches developed in those references, they found that FIGAERO I-CIMS agreed well with AMS-measured mass. It may be worth nothing that assumption (a) is therefore not only probably negligible (per the next lines), but reasonably well supported.

Page 17 line 30: What does "attempt frequency" mean?

Page 23 line 1: missing "are"

Page 23 line 16: Sentence has a typo, not sure exactly what was intended

Caption Figure 4: I think this all refers to Figure 3 panel D, not panel C as stated. Also the descriptions of panels B and C seem to be reversed

Figure 7: Panels C and D are reversed of their descriptions.

Caption Figure 9: typos in lines 1 and 2

Figure 13: I find this figure just generally difficult to interpret, and the plots are very busy with subtly differences between lines. It could uses some re-design.

---

## Referee Comment (RC3) · Anonymous Referee #3 · 18 Jun 2018

The authors present the development of a new model framework with the aim of reproducing thermograms of individual ions, originating from a FIGAERO-CIMS. The model is capable of reproducing the desorption of organic compounds during controlled heating of the filter, by including vapor-surface interactions with the PTFE surfaces, non-idealities from efficient filter heating, together with possible oligomer dissociation and formation processes. Application of this model is performed for calibration experiments and applied to SOA oxidation products originating from the ozonolysis of $\alpha$-pinene. The optimum model fits, possible implications, model simplifications and challenges

are discussed in detail by the authors. This work provides valuable, new insights into the thermodynamic and kinetic properties of SOA using FIGAERO-CIMS, an evolving and active area of research in the field. This publication is suitable for ACP. My suggestions below are mainly to clarify the context and presentation of the results.

Specific comments

I understand that the focus of this study is on the development of a model framework, nevertheless, it would further improve the manuscript if more information regarding the experimental setup/data/uncertainties can be provided, especially since the model is evaluated based on these experimental results. For example, the authors cite in section 2.3 their previous publications and only briefly discuss the experimental details. More information concerning the consistency of the calibrations performed together with their uncertainty would be essential before comparing to the model. Since calibrations and experiments range from 2014 to 2016 it would be informative to discuss the performance of the instrument in these years and how possible changes in the performance could affect the presented results. For example, was the CIMS operated in the same conditions during the calibrations and the chamber studies? These uncertainties should also be included and/or discussed where experimental results are provided (Fig. 3, Fig. 8 - Fig. 12). Since the chamber was operated at steady-state, thermograms throughout the experiment should be consistent and thus an average thermogram should be provided for comparisons to the theoretical approach together with the standard deviation of this average and not just one random thermogram. Finally, information regarding the loadings on the filters and how often the filters were changed would be of value. Were there, for example, any memory/matrix effects observed?

The authors do a nice job of introducing the different type of thermograms and different ways to improve and define the appropriate model fit. However, there is no discussion regarding the model bulk information, although the authors give the impression that this analysis has been already performed. What is the volatility distribution of the

OA mass measured from the FIGAERO based on the model? How does that compare to other experimental results that focus on the volatility of the a-pinene SOA, e.g. Isaacman-VanWertz et al. (2017), or previous model approaches (Lopez-Hilfiker et al., 2014). I consider that these comparisons will be very informative and will further support the evaluation of this model. At the end of the manuscript, the authors suggest that the application of the model will be described in an upcoming publication in more detail. Nevertheless, for the given manuscript the model evaluation could be extended to further promote its capabilities.

Page 7, line 1-10: The assumption used in the model is nicely discussed but it would be beneficial if a rough range of upper mass loadings for the different FIGAERO sampling geometries was provided. For which collection concentration does this uncertainty overcome the model assumption?

Page 11, line 1-2: The decomposition of a compound observed as an oligomer from FIGAERO to lower m/z's is very likely to happen too. How much uncertainty is added to the model due to this assumption? What is the percent of possible oligomers that FIGAERO is able to directly detect in comparison to its total signal?

Page 12, line 5: Add description and modifications in SI

Section 3.6: It will be very informative if the authors add the 8 differential equations in the SI. Characteristic examples of the changes that are applied to the model when more than a single compound is included or the deactivation of certain simplifications could be provided in addition.

Section 3.7: The model computational costs are low for one or two compounds. Let's assume that 100 ions are the main contributors of the OA mass in an SOA experiment; what would then be the computational time for the analysis of all ions when running the model for ideal and non-ideal heating? What is considered high computational costs?

Page 13, lines 21-22: Both Fig. 5 and Table 1 provide information for the model alone

and no experimental results. Although the authors make a comparison of the model to experimental results in Fig. 3 more clear comparison should be provided. An additional column in table 1 with the experimental Tmax from different studies and/or an additional Figure of C* vs Tmax for experimental and modeled, modeled with surface interactions and modeled including non-idealities from efficient filter heating, would directly show whether the model reproduces the Tmax-C* relationship in general.

Page 14, line 1: Delete "less than". For 150 nm particles, the difference between Tmax when excluding and including vapor-surface interactions is very similar to the Tmax difference when changing one volatility bin. This would mean that the underestimation should be around one order of magnitude and not less.

Page 16, line 14: No bulk behavior information is provided in this work.

Page 16, line 15-20: Experimental uncertainties should be added and discussed. See comment above.

Page 16, line 25-26: Provide more details in the SI of how the equations were modified in order to include ammonium sulfate particles.

Page 25, line 12: Define "many". Distribution of the thermograms to the different categories could be provided in more detail.

Technical comments

Page 6, equation (1): For better guidance for the reader it would be nice if the parameters of equation (1) are explained from left to right. This means rearranging the parameters in the equation or/and their explanation on page 6, line 10 to page 7, line 10.

Page 3, line 12: Since the PTR-MS is included as a separate ionization technique, compared to CIMS, a proper citation should be added. An overview of PTR techniques to measure organic aerosol is given by Gkatzelis et al. (2018).

Interactive
comment

Page 3, line 23: Citations are repeated.

Page 3, line 27: Starting a sentence with "But" sounds odd. Maybe rephrase.

Page 5, line 4: "...for a vast majority..."

Page 9, line 11: For clarity the authors should add a sentence of how the model runs were performed, e.g. running equations (1) parallel to (5), and relate this to Fig. 1.

Page 10, line 8: Section 3.4 has the same name as section 3.3.

Section 3.4: There is no consistency between equation numbers (equation (10a) and (10b)), and text (referred as equation (10)).

Page 12, line 29: correct to "Eq. (10a)"

Page 14, line 31: Delete ":"

Page 16, line 4: Delete "a"

Page 21, line 6: Further explanations regarding the different conditions should be provided. What is the RH during these experiments? What is the expected phase-state of the particles?

Page 23, line 1: Missing "are"

Figure 3: These figures are informative but hard to follow. I would recommend that the authors add an additional figure on the right of each panel that represents: x-axis: (temperature of peak desorption)modeled - (temperature of peak desorption)experiment, y-axis: (Full width at half maximum)modeled - (Full width at half maximum)experiment, color: Indicator of the compound as already given from the annotation. This way the difference between modeled and experimental results will show up clearly while the left panels will still be informative regarding the tailing observed.

Figure 4: Panel B and C should be the other way around.

Figure 6: I recommend that the default parameters normalized model thermogram

(C*0=0.1 ug/m3, a=1 etc.) is indicated in all graphs as a dash, bold line and explained in the caption. This way the reader will have a common reference for all cases studied.

Figure 8, 9 and 10: Change the color for high and low volatility for Panel D.

Figure 11 and Figure 12: The colors are not consistent.

References

Gkatzelis, G. I., Tillmann, R., Hohaus, T., Müller, M., Eichler, P., Xu, K.-M., Schlag, P., Schmitt, S. H., Wegener, R., Kaminski, M., Holzinger, R., Wisthaler, A., and Kiendler-Scharr, A.: Comparison of three aerosol chemical characterization techniques utilizing PTR-ToF-MS: a study on freshly formed and aged biogenic SOA, Atmospheric Measurement Techniques, 11, 1481-1500, 10.5194/amt-11-1481-2018, 2018.

Isaacman-VanWertz, G., Massoli, P., O'Brien, R. E., Nowak, J. B., Canagaratna, M. R., Jayne, J. T., Worsnop, D. R., Su, L., Knopf, D. A., Misztal, P. K., Arata, C., Goldstein, A. H., and Kroll, J. H.: Using advanced mass spectrometry techniques to fully characterize atmospheric organic carbon: current capabilities and remaining gaps, Faraday Discussions, 200, 579-598, 10.1039/C7FD00021A, 2017.

Lopez-Hilfiker, F. D., Mohr, C., Ehn, M., Rubach, F., Kleist, E., Wildt, J., Mentel, T. F., Lutz, A., Hallquist, M., Worsnop, D., and Thornton, J. A.: A novel method for online analysis of gas and particle composition: description and evaluation of a Filter Inlet for Gases and AEROsols (FIGAERO), Atmos. Meas. Tech., 7, 983-1001, 10.5194/amt-7-983-2014, 2014.
* * *

---

## Author Comment (AC1) · 10 Aug 2018

**Response to reviewers' comments (received June 12-18, 2018)**
**July 30, 2018**

Title: "A model framework to retrieve thermodynamic and kinetic properties of organic aerosol from composition-resolved thermal desorption measurements"
Paper acp-2018-398
Authors: S. Schobesberger, E. L. D'Ambro, F. D. Lopez-Hilfiker, C. Mohr, and J. A. Thornton

We thank the reviewers for their careful study of our manuscript and their comments. Our point-by-point replies are given below (blue Times New Roman font) following each of the reviewers' comments, which are repeated in full (black Arial font). Reproduced text from the revised manuscript is set in black and green bold Calibri font, green marking changes or additions. New and updated figures are inserted at the end of this document.
* * *
Reviewer #1:

The manuscript by Schobesberger et al. entitled, "A model framework to retrieve thermodynamic and kinetic properties of organic aerosol from composition-resolved thermal desorption measurements" is a wonderful addition to the discussion on the thermodynamic and kinetic properties of secondary organic aerosol. The model framework is described in detail and the analysis of how each fit parameter affects the shape of the desorption curve is useful. Since the best model fit was achieved by characterizing the SOA as being mostly composed of oligomeric-like molecules with parameters that agree well with previous studies, the current manuscript is moving the community closer to a robust set of parameters for describing the thermodynamic and kinetic properties of SOA.

We thank the reviewer for this overall very positive assessment of our manuscript.

Minor revisions suggested: Page Line 6 – "related measures of volatility" are mentioned. Could the authors expand on this?

There may have been a misunderstanding due to a possibly ambiguous formulation. The list item "related measures of volatility" referred to the pair of measures "saturation vapor pressure ($P^*$)" and "saturation vapor concentration ($C^*$)", which are measures of volatility related to each other (and directly proportional at constant temperature). We did not mean to refer to any other measures. To remove any ambiguity, we slightly reformulate the sentence:
**[…] which is primarily controlled by the volatility of the involved compounds, usually expressed as either saturation vapor pressure ($P^*$) or saturation vapor concentration ($C^*$) […]**

In Section 3.7 running a distribution of particle sizes is said to be too computationally expensive. Since this is the case, I would suggest size selecting particles during the experiment so that this source of uncertainty is minimized.

We thank for this suggestion. Indeed, we have since proceeded to include experiments with monodisperse aerosol (size-selected particles), in particular for calibration experiments. Data evaluation is work in progress.

Page 16 line 23 – what was the geometric standard deviation of the particle distribution?

Added that information:
**[…] while the volume median diameter was 197 nm (geometric standard deviation of the volume size distribution = 1.5).**

Page 18 line 16-17: The sentence would read better if "of which" was after the "," and before "the relatively slow decomposition" Page 18 line 19: needs to be "(see also section 4.5)" Page 20 line 3: should be "We used the model" Page 21 line 5 (Section 5.3): Could you please restate the initial values obtained with alpha = 1? Page 23 line 1: should be "convincing and suggests that we are able to attribute" Figure 4: The plots are mislabeled: C should be B and B should be C. The summed trace in what is currently plot 4b is truncated, please give the maximum value of the curve in the caption. Figure 8: should be "A particle size of 197 nm corresponds to the volume median diameter"

Agreed and all changed/added as proposed.

Figure 9: It would be good to keep the colors consistent between Figures 8 and 9. There are numerous typos, please proofread. What does "N.B." stand for? Add that the black traces are non-volatile to the legend for panel D.

Typos have now been hopefully corrected. "N.B." stood for nota bene, but replaced now with simply "Note". The colors between Figs. 8 and 9 (now 9 and 10) should already be consistent. To add some clarity, the legends of panels D were modified to be more descriptive in Figs. 8-10 (now 9-11).

1. Does the paper address relevant scientific questions within the scope of ACP? Yes, the paper discusses experiments and a model framework that aim to further our understanding of the mechanisms of particle evaporation. 2. Does the paper present novel concepts, ideas, tools, or data? Yes, the model framework, supporting experiments, and uncertainty analysis are all novel contributions. The main experiment, the analysis of SOA formed from the dark ozonolysis of alpha-pinene with FIGAERO CIMS, is an important expansion on previous experiments. 3. Are substantial conclusions reached? The quantification of volatility parameters using the model fits to experimental data are substantial conclusions of this manuscript. 4. Are the scientific methods and assumptions valid and clearly outlined? The model framework and all assumptions and sources of uncertainty are described in great detail. 5. Are the results sufficient to support the interpretations and conclusions? The model runs clearly show good agreement with the experimental results and thus support the conclusion that the model parameters describe the thermodynamic and kinetic properties of the system. 6. Is the description of experiments and calculations sufficiently complete and precise to allow their reproduction by fellow scientists (traceability of results)? Yes, the descriptions are all very thorough. 7. Do the authors give proper credit to related work and clearly indicate their own new/original contribution? Yes, previous work is cited where needed and compared to current results. 8. Does the title clearly reflect the contents of the paper? Yes 9. Does the abstract provide a concise and complete summary? The

abstract is a little heavy on describing the methods. More of the abstract could be devoted to the conclusions and best-fit model parameters.

We added a sentence in the abstract, summarizing a selection of the main conclusions from applying the model to a-pinene SOA:

We then discuss the ability of the model to describe thermograms from simple calibration experiments and from complex SOA, and the associated implications for the chemical and physical properties of the SOA. For major individual compositions observed in our SOA test case (#C = 8 to 10), the thermogram peaks can typically be described by assigning $C^*_{25°C}$ values in the range 0.05 to 5 µg m$^{-3}$, leaving the larger, high-temperature fractions (>50%) of the thermograms to be described by thermal decomposition, with dissociation rates on the order of ~ 1 hr$^{-1}$ at 25 °C. We conclude with specific experimental designs […]

10. Is the overall presentation well structured and clear? Yes 11. Is the language fluent and precise? Yes 12. Are mathematical formulae, symbols, abbreviations, and units correctly defined and used? Yes 13. Should any parts of the paper (text, formulae, figures, tables) be clarified, reduced, combined, or eliminated? Yes, there is no explanation for the color scheme in Figure 4. Additionally, the panels are mislabeled in Figure 4.

The panel labels in Fig. 4 have now been corrected (see above), and we also added a sentence in the caption explaining the color scheme:

In all panels, the color scheme reflects the maximum desorption temperature for each fraction or profile, from 200 °C (lightest yellow) to 79 °C (darkest blue in panels B-C) or 25 °C (darkest blue in panel A).

The traces in Figure 8D all appear to be the same color while a legend indicates there should be different colors based on volatility. Figure 9D should have the same color scheme as Figure 8D. The caption for Figure 9 does not read well; it seems as though text became jumbled.

Color legends have been modified to clarify, as mentioned above. The caption of Fig. 9 (now 10) has become jumbled indeed, and is now unjumbled.

Figure 13 is difficult to interpret and needs some more explanation. In general, the axes need to be adjust to show the top portion of the graph, or information given in the caption as to the maximum value when it cannot be seen.

We simplified and redid Fig. 13 (now 14), and tried to make the caption correspondingly clearer. (Plus slight adjustments to the main text as required.)

14. Are the number and quality of references appropriate? Yes 15. Is the amount and quality of supplementary material appropriate? Yes
* * *
Reviewer #2:

The authors present a detailed approach to modeling thermal desorption from a PTFE filter as is found in the FIGAERO CIMS inlet. This model is itself useful in the field, as that instrument is seeing wide use, and the authors further go on to explore properties of SOA, particularly with respect to volatility, kinetics, and potential thermal decomposition. The manuscript is well written and thorough, and represents a clear advance of knowledge. I recommend publication after addressing the comments below, which are largely minor and technical in nature.

Broader comments:
I appreciate the explicit discussion of what is (and is not) meant by "oligomerization" throughout this manuscript. Quite often this term is thought of (perhaps incorrectly, but nevertheless) as implying covalent homo-oligomers, such as dimers and trimers, while there is some evidence and reason to believe that SOA is substantially more complicated than that (e.g. methods able to measure dimers often do not see enough to explain partitioning). While the term is used throughout, this clarity is not brought forward until page 10. I recomment that to address the above conceptions the discussion of "oligomer" be brought forward to the introduction. It is also the reason I tend to prefer "accretion products" as the more universal term, but understand if the authors prefer to stick with the more common term "oligomers".

Agreed about the superiority, in that sense, of the term "accretion products", but we do stick to "oligomers", because, as stated, it appears to be more commonly used in the field to refer to accretion products in SOA, and also because it is a bit shorter.
It is a good suggestion to bring up our explicit definition of "oligomer" already in the introduction, specifically we do that now in page 3:
**Speculations have included ubiquitous peroxides (cf., Docherty et al., 2005) with breakage of the O–O bond upon heating, networks of H-bridge bonds in the SOA matrix that are stronger or denser than for pure compounds or ideal mixtures, and oligomeric structures initially in thermodynamic equilibrium with monomers and thus dissociating during heating to re-achieve equilibrium (Lopez-Hilfiker et al., 2015). Consequently, we are using a broad and inclusive definition of the term "oligomer" in this study, referring to any physical entity that is essentially non-volatile but incorporates and/or releases generally more volatile molecules (the latter in particular upon heating). I.e., our definition is considerably more universal than the frequent use of the term as referring specifically to covalently bound large molecular weight molecules.**
The original discussion of the term (page 10) is correspondingly slightly modified.

Throughout the manuscript, the authors sometimes mention the possibility that an ion may represent multiple isomers, but its not always clear to me to what extent this is being considered. Figure 11 demonstrates that there are many possible ways to fit each thermogram, and one could imagine for instance an ion consisting of monomers with a range of volatilies, and each also formed from one or two different oligomers. Panel B shows such an example a case in which two isomers are invoked to describe the thermogram, one which is pure and another comprised of low- and high-volatility components. While captured here, it may be a case existing in many of the observed ions, but to which the model is blind given its large number of free parameters. It seems here that in most cases there is a simplifying assumption that each ion can be treated as one compound (except in the case of 11B), which may or may not be a robust assumption. Trying to tackle this question may go beyond the scope of this manuscript, but it should probably be discussed more explicitly and added to the list of "challenges" in Section 5.

The reviewer is entirely correct, and thinking about this issue again, it, and particular its implications, may not have been adequately discussed. In general, in this study (except for Figs. 8 and 12C, now 9 and 13C), we attempted to use the smallest possible number of isomers, and almost always just a single isomer sufficed. Only in the case illustrated by Fig. 11B (now 12B), the assumption of two isomers was essentially required due to the complex shape of the measured thermogram. The likely effect of multiple isomers, even if technically not required, is actually shown in Fig. 12 (now 13; cf. panels A and C) and Table 2. The primary effect is that overall lower saturation concentrations ($C^*_0$) and higher vaporization enthalpies ($\Delta H$) need to be used to simulate observations, if there are isomers differing in $C^*_0$. As suggested, we include a corresponding discussion in section 5.6:

**Another issue, worth pointing out again, is the possible errors introduced if there are indeed multiple isomers contributing to a single composition's thermogram, if their volatilities ($C^*_0$) differ, but not by enough to be revealed by separate thermogram peaks. We show a possible ambiguity of this type for $C_{18}H_{28}O_6$ (cf. Fig. 13A and 13C; Table 2). The primary effect of simulating an observation of a single thermogram peak by assuming multiple isomers (i.e., multiple $C^*_0$), instead of a single isomer, is that overall lower $C^*_0$ and higher $\Delta H$ need to be used.**

Technical comments:
Page 3 line 10: A word seems to be missing in "Other methods by"
Corrected.

Page 3 line 17: What do the author's mean by the "compositions of these molecules"? I presume they mean molecular formulas?
Yes, corrected.

Page 10 lines 18-22:
Page 10 line 27: The use of "compound" here and throughough is 'iffy', as for most practical FIGAERO applications a given ion may represent a mixture of multiple compounds. See comment above.
That is correct. Our excessive use of the term "compound" probably owes to the primary drafting author not being a chemist by training, maybe compounded by not having English as a native language. We added a clarifying sentence at the end of section 2.1 ("FIGAERO-CIMS"):
**Note that the CIMS can measure only elemental compositions, i.e. molecular formulas. Consequently, the identities of the specific compounds remain ambiguous in general.**
Throughout the text, we replaced "compound" with "composition" where appropriate. Mostly, that is in regards to CIMS measurements relating to certain molecular formulas or ions.

Page 11 line 18: It's not clear to me how k_g,0/(k_g,0+k_d,0) is calculated before each model run
We hope to make it clearer by changing two sentences at that place to:
**The fraction of molecules initially present in the oligomer state was then simply $k_{g,0}/(k_{g,0}+k_{d,0})$. This fraction was  used as an initial condition for $N_g$.**
Maybe the confusion was due to "before each model run": by "model run" here was meant specifically the solving of the differential equations. The free parameters are chosen in the model before the equations are solved, therefore that ratio (= $N_g$ at t = 0) can be calculated as well.

Page 12 line 7-9: My understanding of both cited references is that FIGAERO CIMS saw half the mass, but also assumed equal sensitivity. However, when Isaacman- VanWertz et al. (Nature Chemistry, 2018, doi:10.1038/s41557-018-0002-2) applied the calibration approaches developed in those references, they found that FIGAERO I-CIMS agreed well with AMS-measured mass. It may be worth nothing that assumption (a) is therefore not only probably negligible (per the next lines), but reasonably well supported.

Thanks for pointing out the FIGAERO measurements for the Isaacman-VanWertz et al. paper, and their encouraging results regarding agreement with the AMS when using a calibration approach clearly superior to simply assuming a single sensitivity. We incorporated that information in a slightly modified last paragraph of section 3.5:

We know that FIGAERO coupled to iodide-CIMS appears to detect only about half of the organic material by mass under these assumptions, and that reported sensitivities generally vary widely (Lopez-Hilfiker et al., 2016b; Iyer et al., 2016). However,  even if only half of the organic mass was accounted for, the directly introduced error would be comparable with an error in $C^*_i$ or $\alpha$ of up to about a factor of two, which will be a relatively small uncertainty given other ambiguities discussed below. Indeed, a recent study employed a calibration procedure for instrument sensitivity to most compositions and, within uncertainties, obtained mass closure with independent AMS or SMPS measurements, lending support to assumption (a) (Isaacman-VanWertz et al., 2017; Isaacman-VanWertz et al., 2018).  Assumption (b) may introduce bigger errors, particularly if sensitivity to compound i is far from the average, though we argue  these errors are generally smaller for compounds that desorb at higher temperatures […]

Page 17 line 30: What does "attempt frequency" mean?
The term refers to the pre-exponential factor in the Arrhenius equation. We now simply call it "pre-exponential factor", which is probably more generally used.

Page 23 line 1: missing "are"
Corrected.

Page 23 line 16: Sentence has a typo, not sure exactly what was intended
No typo detected, but added a few words to make the sentence clearer:
Stark et al. (2017) also pointed out that there can be large differences in the response of individual FIGAERO instruments during calibration experiments, in particular  regarding $T_{max}$, and in particular when differences in the exact instrument designs are involved.

Caption Figure 4: I think this all refers to Figure 3 panel D, not panel C as stated. Also the descriptions of panels B and C seem to be reversed
Correct. The caption and figure have been fixed now.
Figure 7: Panels C and D are reversed of their descriptions.
Caption Figure 9: typos in lines 1 and 2
Both corrected. (Special thanks at this point for the careful read!)

Figure 13: I find this figure just generally difficult to interpret, and the plots are very busy with subtly differences between lines. It could uses some re-design.
See also comment by Reviewer #1. We simplified and restructured Fig. 13 (now 14). We hope that it is much clearer now and easier to understand.

\*\*\*\*\*\*\*\*\*\*\*\*\*\*\*\*\*\*\*\*\*\*

Reviewer #3:

The authors present the development of a new model framework with the aim of reproducing thermograms of individual ions, originating from a FIGAERO-CIMS. The model is capable of reproducing the desorption of organic compounds during controlled heating of the filter, by including vapor-surface interactions with the PTFE surfaces, non-idealities from efficient filter heating, together with possible oligomer dissociation and formation processes. Application of this model is performed for calibration experiments and applied to SOA oxidation products originating from the ozonolysis of α-pinene. The optimum model fits, possible implications, model simplifications and challenges are discussed in detail by the authors. This work provides valuable, new insights into the thermodynamic and kinetic properties of SOA using FIGAERO-CIMS, an evolving and active area of research in the field. This publication is suitable for ACP. My suggestions below are mainly to clarify the context and presentation of the results.

Specific comments

I understand that the focus of this study is on the development of a model framework, nevertheless, it would further improve the manuscript if more information regarding the experimental setup/data/uncertainties can be provided, especially since the model is evaluated based on these experimental results. For example, the authors cite in section 2.3 their previous publications and only briefly discuss the experimental details.

We moved up some experimental setup information from section 5 to a probably more appropriate location in section 2.3 (last paragraph). We believe that a more thorough discussion is beyond the scope of the already rather lengthy manuscript, and kindly refer to the cited previous works to obtain a fuller picture if desired.

More information concerning the consistency of the calibrations performed together with their uncertainty would be essential before comparing to the model. Since calibrations and experiments range from 2014 to 2016 it would be informative to discuss the performance of the instrument in these years and how possible changes in the performance could affect the presented results. For example, was the CIMS operated in the same conditions during the calibrations and the chamber studies?

The used instrument, including the FIGAERO inlet, was the same in all experiments used in this work. Certainly, certain instrument parameters change over time, or with conditions, affecting for instance sensitivity. As our work here focuses on the interpretation of the overall shape of the thermograms of individual compositions, shifts in sensitivity are not an issue. We expect the thermogram shapes (including, e.g., the often-used measurement of $T_{max}$) to be very consistent over time, provided that the sampling and thermal desorption geometries do not change, in particular the heater setup and thermocouple position, which effectively control how the measured desorption temperature profile relates to the actual temperatures (ideally identical). To our knowledge, no such changes have occurred between the calibration and SOA experiments

used in this study. But the accuracy, and ultimately applicability, of the model clearly strongly depend on such instrumental stability.

Added the following sentence into section 5.6:

**It appears likely that model re-calibration is also necessary whenever the sampling or thermal desorption geometry of a specific instrument has changed, in particular the heater setup including the position of the thermocouples used for measuring the desorption temperature profile.**

See also our reply below, regarding uncertainties.

These uncertainties should also be included and/or discussed where experimental results are provided (Fig. 3, Fig. 8 - Fig. 12). Since the chamber was operated at steady-state, thermograms throughout the experiment should be consistent and thus an average thermogram should be provided for comparisons to the theoretical approach together with the standard deviation of this average and not just one random thermogram.

This is a good suggestion, but there are general difficulties or caveats involved with obtaining such uncertainties. For Fig. 3, experimental data was used that is presented in previously published work (Lopez-Hilfiker et al., 2014), and additional data for deriving uncertainties is now difficult to retrieve, in particular for the calibration experiment data used in Fig. 3. However, that work does include a discussion and data on reproducibility and variability of thermograms for individual compositions from SOA chamber experiments. Therein, variability and reproducibility of absolute signal is attributed mostly to variations in the blank (i.e. background) measurements, introducing an uncertainty of about 5%. A good reproducibility of thermograms was also shown e.g. in Huang et al. (2018).

Our datasets for the SOA experiments here generally do not allow for deriving our own value, even though we operated as continuous flow reactor. That is because we did not dwell long enough on any given steady state, due to practical time constraints. The FIGAERO was the slowest instrument to analyze any given conditions, so usually we only took enough data for obtaining a single thermogram once steady-state conditions were reached. (A single standard FIGAERO measurement required 40 min of sampling from the chamber plus 80 min for the full desorption cycle. A blank measurement took the same amount of time again.)

Nonetheless, we took a more careful look into the time evolution of the thermograms for our test case, and now include in the supplement a collection of sequentially taken thermograms for $C_8H_{12}O_5$ (the composition with the leading role in this paper, Figs. 8-10, now 9-11) leading up to the steady-state chamber conditions (Fig. S1). The figure shows, as in previous works, that the thermograms are actually remarkably reproducible. Most importantly, the thermogram *shape* is particularly stable with time (center panels), even if steady-state has not yet been reached, and also for the blank-corrected thermograms, which are subject to the additional variation between the blank measurements. The bottom panels show average normalized thermograms from the final three FIGAERO measurements, i.e. as steady-state conditions were approached in the chamber, which is practically indistinguishable from the final measurement, i.e. from the one used for the thorough analysis in the manuscript (Figs. 8-10, now 9-11). Therefore, we keep the single experimental thermogram in Figs. 8-12 (now 9-13), for lack of sufficient statistics on one hand, but justified by the high reproducibility of individual thermograms.

Besides the section in the supplemental material, we added to the first paragraph of section 5.1:

**Our measured thermogram shapes for a given chamber condition were highly reproducible (supplemental material, Fig. S1), as expected from previous studies (Lopez-Hilfiker et al., 2014), and we therefore neglect experimental uncertainties in the following. However, we generally do expect changes in thermogram shapes for individual compositions, if there are changes in the instrumental setup (section 5.6) or experimental conditions.**

Finally, information regarding the loadings on the filters and how often the filters were changed would be of value. Were there, for example, any memory/matrix effects observed?

Added the following sentence to the introductory part of section 5.1:
**The calculated filter loading for this experiment was 0.31 µg. No matrix effects were apparent.**
Also added at the end of section 2.3 ("Experiment setups"):
**Typical SOA mass loadings in the chamber were 2 to 3 µg m$^{-3}$, and the FIGAERO achieved adequate filter loadings by sampling for 40 min periods at 2.5 L min$^{-1}$. Every 4$^{th}$ sample was a blank measurement, with an additional filter in the aerosol sampling line (Lopez-Hilfiker et al., 2014). Measurement results were continuously monitored and both filters were replaced when memory effects in the form of elevated backgrounds were noticed (on average once per week).**

The authors do a nice job of introducing the different type of thermograms and differentways to improve and define the appropriate model fit. However, there is no discussion regarding the model bulk information, although the authors give the impression that this analysis has been already performed. What is the volatility distribution of the OA mass measured from the FIGAERO based on the model?

The model bulk information for these experiments has actually not been retrieved, at least not to the same quality as done for our selection of individual compounds. For evaluating Eqs. 14-15, it is indeed necessary to assign at least some $C^*_0$ and $\Delta H$ values to the bulk OA mass so that the bulk thermogram can be simulated, but the only purpose of that is to obtain $N_R$ as a function of time (and hence a rough reproduction of the experimental data turns out to be sufficient). However, those $C^*_0$ and $\Delta H$ values (and consequently other free model parameters affecting the bulk thermogram) turn out to be physically rather meaningless, because the bulk thermogram is a superposition of the thermogram signals of all individual compositions, which we know differ substantially in their respective volatilities. Therefore, any single pair of $C^*_0$ and $\Delta H$ used to fit the bulk thermogram will consistently yield an overestimate of the actual average $C^*_0$ and an underestimate of the average $\Delta H$. We actually demonstrate this effect in Fig. 12 (now 13; cf. panels A and C) and its discussion at the end of section 5.2, as the thermogram for the composition $C_{18}H_{28}O_6$ is probably best explained by the superposition of 3 isomers, whereas its explanation by a single isomer yields an unexpectedly high $C^*_0$ combined with an unreasonably low $\Delta H$. (Following a comment on the effect of isomers by reviewer 2, this ambiguity is now pointed out explicitly in section 5.6.)
Consequently, the proper way to obtained bulk information would be to analyze the thermograms for each individual composition, at least those substantially contributing to the total. Such analysis has not yet been done, but hopefully more feasible in the future, and we are currently developing automatic fitting routines to help facilitate it. Alternatively (or additionally), experiments can be designed to include isothermal evaporation phases (such as proposed in

section 5.7), that allow for additional constraints by decoupling $C^*_0$ at least from $\Delta H$, and may thus be helpful for retrieving bulk OA information.

To better clarify what we did to obtain $N_R(t)$, we added in the 1st paragraph in section 3.5:

**The latter sum is treated like a single composition by the model, and the respective model parameters may be unphysical,  because the corresponding sum thermogram is a superposition of the thermogram signals of all individual compositions, which we know differ substantially in their respective volatilities. Nonetheless, the parameters are chosen such that the corresponding thermogram is adequately reproduced and thus allow us to use appropriate values for $\chi_i$, $D_P$ and $\Phi$ as functions of time.**

How does that compare to other experimental results that focus on the volatility of the a-pinene SOA, e.g. Isaacman-VanWertz et al. (2017), or previous model approaches (Lopez-Hilfiker et al., 2014). I consider that these comparisons will be very informative and will further support the evaluation of this model. At the end of the manuscript, the authors suggest that the application of the model will be described in an upcoming publication in more detail. Nevertheless, for the given manuscript the model evaluation could be extended to further promote its capabilities.

We completely agree that the analysis of the "bulk" SOA (or, in practice, the major compositions) using our model will be very interesting, in particular also comparisons with previous works. With the current model version being made public, and our colleagues working on automated parameter finding routines (via optimization algorithms), we hope to follow up soon with such an analysis work. It is understood that the proposed comparisons would already help promoting the model. Even more helpful, however, may be additional application or comparison to calibration experiments, potentially using other FIGAERO instruments…

But as the manuscript is already lengthy (which we think is OK for a paper introducing a new model framework), we rather refrain from extending it by including further analysis at this time and kindly refer to those future publications.

Page 7, line 1-10: The assumption used in the model is nicely discussed but it would be beneficial if a rough range of upper mass loadings for the different FIGAERO sampling geometries was provided. For which collection concentration does this uncertainty overcome the model assumption?

Unfortunately, we feel that we do not have sufficient information to provide such an upper limit mass loading with the necessary confidence. But let that not stop us from trying anyway here:

The size of the filter area onto which particles are deposited may be visually assessable following deposition of enough material to optically discolor the filter. For the sampling geometry used in our work here (the UW design), the 24-mm filter did appear to be loaded uniformly (i.e. it becomes visibly dirty throughout, after many days of experiments, except for the very edge that is covered by the filter holder), so that the coverage by an OA deposit of <0.3 µg (200 nm particles) was likely <1%. Huang et al. (2018), however, observed matrix effects starting already at 0.5 µg. They did have a different sampling geometry, the Aerodyne design, observed by some of us to focus the particles onto a smaller spot than the UW design, roughly ¼" wide. With that information, one may conclude that in order to try to avoid matrix effects, specifically by keeping coverage at <10%, filter loadings should be kept below ca. 0.5 µg for the Aerodyne design, whereas below ca. 5 µg would suffice for the UW design.

However, much of the information used here is rather anecdotal. And an additional unknown, in principle, is where exactly in the filter do the particles actually deposit. For example, deposition may occur preferentially at certain "hotspots" defined by the detailed (microscopic-scale) interactions between non-homogeneous filter material, air flow and aerosol particles, thus enhancing matrix effects. Clearly more rigorous experimentation is desirable to properly explore (assumed) observations of matrix effects for various filter loadings and sampling setups. But at the moment, we do not feel qualified to make statements beyond what is currently in the manuscript.

Page 11, line 1-2: The decomposition of a compound observed as an oligomer from FIGAERO to lower m/z's is very likely to happen too. How much uncertainty is added to the model due to this assumption? What is the percent of possible oligomers that FIGAERO is able to directly detect in comparison to its total signal?

In short, we do not know, and we have not made the effort (yet?) to add this possible type of decomposition to the model. We are therefore also unable to assess the requested uncertainty and fraction. However, the thermogram shapes observed when desorbing SOA can be explained without the addition of decomposition of the respectively observed compositions. A possible conclusion is that this kind of decomposition is negligible. For some cases, (e.g. citric acid, possibly some organic nitrates in ambient data), thermograms have steeper drop-offs towards higher desorption temperatures than predicted by the model, most likely due to such decomposition, and such data could be explored in future work. The issue is briefly discussed in sections 3.4 and 5.6

Page 12, line 5: Add description and modifications in SI
Added a supplement with that information, plus reference in the main text.

Section 3.6: It will be very informative if the authors add the 8 differential equations in the SI. Characteristic examples of the changes that are applied to the model when more than a single compound is included or the deactivation of certain simplifications could be provided in addition.
Added those 8 equations to the supplement, as well as a description of how options affect that set of equations, plus reference in the main text.

Section 3.7: The model computational costs are low for one or two compounds. Let's assume that 100 ions are the main contributors of the OA mass in an SOA experiment; what would then be the computational time for the analysis of all ions when running the model for ideal and non-ideal heating? What is considered high computational costs?
We deliberately avoid speaking about "low" or "high" computational costs, as that appears to be usually a subjective classification. Personally, we have been satisfied if running the model takes at most several seconds on our business- or consumer-range desktop or laptop computers, which turned out to be what it takes to analyze up to a few ions for non-ideal heating. We have not systematically investigated how much computational costs actually increase with adding compounds to the simulation. (The number of equations typically increases linearly, the computational costs presumably somewhat more slowly.) In practice, however, most time is likely spent by optimizing the free parameters for reproducing (fitting to) the observed thermogram(s), which is currently still a manual process that requires to run the model several to many times. Automating this process via some efficient optimization algorithm is probably the

way to go, at first, for making the application of our model for efficient overall. To clarify to the reader, where the problem lies regarding model optimization, we added a paragraph to section 3.7:

**Parameter optimization, i.e. finding the values for the free parameters that reproduce an observed thermogram, is currently still manual, requiring multiple model runs. The number of required runs depends on thermogram complexity and operator experience, 20 to 40 runs being typical. Future steps for making model application more efficient will be automation of that process through optimization algorithms, e.g. genetic algorithms.**

Page 13, lines 21-22: Both Fig. 5 and Table 1 provide information for the model alone and no experimental results. Although the authors make a comparison of the model to experimental results in Fig. 3 more clear comparison should be provided. An additional column in table 1 with the experimental Tmax from different studies and/or an additional Figure of C* vs Tmax for experimental and modeled, modeled with surface interactions and modeled including non-idealities from efficient filter heating, would directly show whether the model reproduces the Tmax-C* relationship in general.

Good point, we also thought such a plot would be useful (but after the Discussions manuscript had already been submitted). We added the new Fig. 7. The figure is announced it at the end of section 4.1:

**In the following sections, we will see how other model input parameters affect $T_{max}$ as well, and revisit in section 4.4. the model reproduction of the $T_{max}$-$C^*$ relationship.**

And it is introduced at the end of section 4.4:

**Figure 7 summarizes how, for otherwise typical assumptions and conditions, the simulated $T_{max}$ is defined by $C^*_0$ and $\Delta H$. (colored line), generalizing the $T_{max}$-$C^*$ relationship found previously based on experimental observations (Lopez-Hilfiker et al., 2014; Mohr et al., 2017; colored circles and black line). As the colors of the circles roughly match those of the underlying model-derived lines, the model largely reproduces the empirical relationship, as seen above (section 4.1, Fig. 3D). Conversely, comparison with the model results infers a relation between $C^*_0$ and $\Delta H$ (colored lines vs. black line), which consistently predicts relatively lower values for $\Delta H$ than an independent semi-empirical $C^*_0$-$\Delta H$ relation (Epstein et al., 2010; black dashed line).**

Page 14, line 1: Delete "less than". For 150 nm particles, the difference between Tmax when excluding and including vapor-surface interactions is very similar to the Tmax difference when changing one volatility bin. This would mean that the underestimation should be around one order of magnitude and not less.

Agreed and changed.

Page 16, line 14: No bulk behavior information is provided in this work.

Agreed and shortened the sentence accordingly. For more details about this lack of information, see reply above.

Page 16, line 15-20: Experimental uncertainties should be added and discussed. See comment above.

See replies to comments above.

Page 16, line 25-26: Provide more details in the SI of how the equations were modified in order to include ammonium sulfate particles.

Done, as mentioned above, and here included a reference to the supplement.

Page 25, line 12: Define "many". Distribution of the thermograms to the different categories could be provided in more detail.

Replaced "many" with "the majority of". We did not systematically analyze the full dataset, so we cannot be more precise at the moment, nor provide requested distribution. But we hope that that kind of analysis will be the subject of future work aided by our model.

Technical comments

Page 6, equation (1): For better guidance for the reader it would be nice if the parameters of equation (1) are explained from left to right. This means rearranging the parameters in the equation or/and their explanation on page 6, line 10 to page 7, line 10.

Rearranged/modified Eq. 1 and the following text as suggested.

Page 3, line 12: Since the PTR-MS is included as a separate ionization technique, compared to CIMS, a proper citation should be added. An overview of PTR techniques to measure organic aerosol is given by Gkatzelis et al. (2018).

Added.

Page 3, line 23: Citations are repeated.

Removed.

Page 3, line 27: Starting a sentence with "But" sounds odd. Maybe rephrase.

Conjugated.

Page 5, line 4: "…for a vast majority…"

Thanks.

Page 9, line 11: For clarity the authors should add a sentence of how the model runs were performed, e.g. running equations (1) parallel to (5), and relate this to Fig. 1.

Added information to supplement.

Page 10, line 8: Section 3.4 has the same name as section 3.3. Section 3.4: There is no consistency between equation numbers (equation (10a) and (10b)), and text (referred as equation (10)).

Thanks for noticing this mistake! Section 3.4 is now appropriately named **Implementation of oligomerization reactions**. Also corrected the references to "Eq. (10)".

Page 12, line 29: correct to "Eq. (10a)"

Page 14, line 31: Delete ":"

Page 16, line 4: Delete "a"

Corrected.

Page 21, line 6: Further explanations regarding the different conditions should be provided. What is the RH during these experiments? What is the expected phase-state of the particles?

The RH is indeed likely to be one of the major differences, where the cited Saleh et al. (2013) used almost dry conditions (<10% prior to heating). We added that information, but cannot really add more, as Saleh et al. themselves were not very specific, unfortunately. (Their SOA loadings and precursor concentrations range widely, including our conditions at the PNNL chamber.)

Page 23, line 1: Missing "are"

Corrected.

Figure 3: These figures are informative but hard to follow. I would recommend that the authors add an additional figure on the right of each panel that represents: x-axis: (temperature of peak desorption)modeled - (temperature of peak desorption)experiment, y-axis: (Full width at half

maximum)modeled - (Full width at half maximum)experiment, color: Indicator of the compound as already given from the annotation. This way the difference between modeled and experimental results will show up clearly while the left panels will still be informative regarding the tailing observed.

Interesting idea, "crosshair plots". Implemented as suggested.

Figure 4: Panel B and C should be the other way around.

Corrected.

Figure 6: I recommend that the default parameters normalized model thermogram (C*0=0.1 ug/m3, a=1 etc.) is indicated in all graphs as a dash, bold line and explained in the caption. This way the reader will have a common reference for all cases studied.

Yes, that might help some readers orient themselves. A very bold gray background line worked well for including the default case. Figure and caption correspondingly updated.

Figure 8, 9 and 10: Change the color for high and low volatility for Panel D.

To add some clarity, the legends of panels D were modified to be more descriptive in Figs. 8-10 (now 9-11). (See also comment by Reviewer 1).

Figure 11 and Figure 12: The colors are not consistent.

It is a somewhat complex color scheme used here, but we did not make out the inconsistency.

References

[revised manuscript text omitted]

**Figure S1: The seven thermograms for composition $C_8H_{12}O_5$ leading up to the thermogram used in this work (section 5.1, Figs. 9-11), which is included in the darkest blue and representing steady-state conditions for dark α-pinene ozonolysis ([$O_3$] = 84 ppbv, [α-pinene reacted] = 6.7 ppbv). The color scheme represents time at which the sample was taken from the chamber. Desorption starts at 0 s. The vertical dashed lines at ca. 1200 s mark the time when a desorption temperature of 200 °C has been reached and is subsequently maintained. (Data are shown only until 2500 s since the start of desorption to show more clearly the time during the temperature ramp (25 to 200 °C), which contains most information.) The top left-hand panel shows count rates adjusted for reagent ion concentration and volume of sampled chamber air; the top right-hand panel is additionally corrected for background signal as determined by blank measurements. In the center panels, the data are normalized to 1 for comparing thermogram shapes. The bottom panels reproduces the final thermogram (dark blue line) and also shows the mean of the final three thermograms (black bold line) plus standard deviation (gray shades).**

---

## Author Comment (AC2) · 16 Aug 2018

As announced in the discussion paper, a documented version of the model's MATLAB code is now available for download. The model is hosted on GitHub and can be publicly accessed at github.com/sschobes/FIGAERO_model.